# Learning to Safely Exploit a Non-Stationary Opponent

## Abstract

In dynamic multi-player games, an effective way to exploit an opponent's weaknesses is to build a perfectly accurate opponent model. This renders the learning problem a single-agent optimization which can be solved by typical reinforcement learning. However, naive behavior cloning may not suffice to train an exploiting policy because opponents' behaviors are often non-stationary due to their adaptations in response to other agents' strategies. On the other hand, overfitting to an opponent (i.e., exploiting only one specific type of opponent) makes the learning player easily exploitable by others. To address the above problems, we propose a method named Exploit Policy-Space Opponent Model (EPSOM). In EPSOM, we model an opponent's non-stationarity as a series of transitions among different policies, and formulate such a transition process through Bayesian non-parametric methods. To account for the trade-off between *exploitation* and *exploitability*, we train a player to learn a robust best response to the opponent's predicted strategy by solving a modified meta-game in policy space. In this work, we consider a two-player zero-sum game setting and evaluate EPSOM on Kuhn poker; results suggest that our method is capable of exploiting its adaptive opponent, whilst maintaining low exploitability (i.e., achieving safe opponent exploitation). Furthermore, we show that our EPSOM agent has strong performance against unknown non-stationary opponents without further training.

## 1  Introduction

In single agent reinforcement learning (SARL), an agent learns to act by iteratively interacting with an environment. In such a setting, an agent's learning objective and its performance evaluation are normally clear and straightforward, e.g., its long-term cumulative rewards gained from the environment. However, in multi-agent reinforcement learning (MARL), one agent's performance greatly depends on the behavior of other agents. Hence, finding a reliable learning objective and evaluation method become non-trivial [3, 9, 31, 48]. Naive solutions of the problem using SARL generalize badly [21] and optimizing the joint policy of all agents does not scale. Recent approaches combining game theoretical analysis with deep RL have seen some success in large zero-sum games [4, 44].

Game theory offers a mathematical framework to model strategic interactions among players [28]. Under perfect rationality [12], a central solution concept is Nash equilibrium (NE) where no player benefits from deviating from their equilibrium strategy. In a two-player zero-sum game without any inherent advantage for either player (e.g. as a first mover), a NE is a safe strategy to play (i.e., playing not to lose) – NE guarantees a tie in the worst case in expectation. However, NE is not the most profitable strategy in many cases. In complex competitive games, such as poker, it is common that agents encounter opponents with bounded rationality, in the sense that they may at best play an

Submitted to 35th Conference on Neural Information Processing Systems (NeurIPS 2021). Do not distribute.

approximate Nash equilibrium strategy and often play dominated actions [5, 33]. Therefore, playing a NE can potentially forego significant rewards against sub-optimal opponents. This incentivizes players to deviate from the NE and exploit their opponents' weakness (i.e., playing to win). However, the resulting strategy could render itself exploitable should it overfit to the current opponent. Playing to win can therefore lead to exploitation by other opponent strategies. In the case of deceptive opponents such exploitation is known as the "get taught and exploited" problem [35].

To better balance the trade-off between playing to win against the current opponents (exploitation) and not losing to unknown opponents (exploitability), Johanson et al. [19] proposed a solution concept, named Restricted Nash Response (RNR). RNR and its variants [18, 19, 33] assume stationary opponents, i.e., the strategies they learn to exploit are unknown but fixed. However, in many real-world applications, opponents may adapt and change their strategies on an ongoing basis. For example, in Rock-Paper-Scissors when a player learns to best respond by playing Rock to an opponent's strategy which always plays Scissors, the opponent may then learn to best respond to your best response by playing Paper. Furthermore, prior RNR approaches only provide one-off solutions in the sense that whenever we need to re-adjust the trade-off between exploitation and exploitability or the opponent uses a new fixed policy, we need to re-solve the updated game from scratch.

In this work, we focus on problems with non-stationary opponents. An opponent's learning process can be generally modeled as transitions among a mixture of unknown number of policies. This motivates the usage of a Dirichlet process mixture model. As we can only collect trajectories produced by the adaptive opponents online, we propose to learn our model in a streaming fashion. Given the predicted opponent policy from our model, we provide a general framework for training an agent to safely exploit the non-stationary opponent where safe exploitation means exploiting the current opponent with a low probability of being exploited by other opponents in future. We empirically demonstrate the ability of our approach to safely exploit a non-stationary opponent in Kuhn Poker, a simplified Poker game. Furthermore, once trained, our model can produce strong counter strategies to unseen opponents without any further training in new tournaments.

## 2 Related Works

A fundamental ability of an effective AI agent is the capacity to interact with other intelligent agents. Therefore, the capability of reasoning about other agents' goals [34], private information [27], behavior [13] and other characteristics is crucial. The issue of non-stationarity in multi-agent systems resulting from coexisting agents is well-known and well documented [14, 32]. Classical solutions to resolve the issue of non-stationarity include centralized training [24], self-play [44], meta-learning [1] and opponent modeling [2]. When specifically applied to the issue of non-stationarity, most previous works focusing on opponent modeling which switches between different opponent models when a change in opponent(s) is detected. A switch of model may be triggered by a drop in opponent model prediction accuracy [10] or when performance in terms of reward received for a fixed policy drops [15]. Deep BPR+ [49] combines a measure of opponent model accuracy and reward tracking to decide when to learn a new policy. Significantly, most of these works limit the opponents' non-stationarity to periodically changing their policies within a finite pre-defined set of stationary policies.

In this work, we consider non-stationarity during the training stage arising from the opponents' concurrent learning dynamics, rather than drawing stationary opponents from a pre-defined set. The entire lifetime of an opponent can generally be modeled as a mixture of an unknown (possibly infinite) number of policies. This motivates the usage of a Dirichlet Process (DP) mixture model [6, 42] which can infer the number of mixture components from data and provide incremental model capacity on demand. Various approximate inference methods are reported for DP mixture models, such as Markov chain Monte Carlo [17] and variational inference [6, 16, 45]. However, these inference methods either do not adapt to an online setting or truncate the number of clusters to a finite value. Recently proposed streaming inference algorithms [23, 41] enable the DP mixture model to solve online non-stationary problems in a truly non-parametric way. Applications have been reported in task-free continual learning [22] and model-based reinforcement learning [47]. In this paper, we adopt this approach to model and simulate a non-stationary opponent for MARL.

It is well known that finding a NE is PPAD-hard even in two-player games [8]. An exception is two-player zero-sum games where the NE can be tractably solved by a linear program (LP) in polynomial time [43]. However, in games with extremely large action spaces, approximate NE solutions, such

as fictitious play (FP) [7] and counterfactual regret minimisation (CFR) [50], have to be used. An important design principle that underpins NE approximation is the iterative best-response dynamics. Two representative methods are Double Oracle (DO) [26] and Policy Space Response Oracle (PSRO) [21]. In the dynamics of DO [26], players are initialized with restricted strategy sets; then at each iteration, a NE will be computed over the current restricted sets. These sets will be expanded by adding the best-response strategy to the NE computed over the full strategy sets. The iterative process continues until the best response is in the restricted strategy pool. PSRO approximates DO by interleaving empirical game-theoretic analysis (EGTA) with deep RL. In contrast with DO, the game with restricted strategy sets has to be estimated through simulation. Furthermore, the exact analytical best-response oracle is replaced in PSRO by a deep RL oracle which calculates an approximate best response. PSRO is a general self-play framework for MARL and many approaches built upon it have been proposed to improve its performance [25, 29, 30, 39]. Our approach, EPSOM, is not limited to the self-play setting. In addition, although we favor solutions with low exploitability (i.e. solutions close to NE) as PSRO does, our ultimate goal is to find a robust best response to a non-stationary opponent rather than solving the game for an (approximate) NE.

# 3 Preliminaries

We consider the decentralized training and decentralized execution (DTDE) setting in zero-sum games where we have access to interaction trajectories $\tau$ between our agent and the opponent[i] . Whilst our approach can be extended to games with multiple opponents, in this work we focus on 2-player zero-sum games. Before introducing our algorithm, we present some necessary preliminary concepts and notation in the remainder of this section.

## 3.1 Meta Normal-Form Game

We consider opponent modeling in policy space and learn to respond to the predicted distribution of the opponents' policies. We formulate this problem as solving a two-player normal-form game (NFG) between our agent and its opponents as a whole with notation adapted to our presentation. We denote a 2-player NFG by a tuple $(\Pi, U, \mathcal{N})$ where $\Pi^i$ is player $i$'s set of policies and $i \in \mathcal{N}$ where $\mathcal{N} = \{1, 2\}$. For ease of notation, we take player 1 as the training agent and player 2 as its opponent. We use $\Pi = \prod_{i \in \mathcal{N}} \Pi^i$ to denote the set of joint policies (strategy profiles). $U(\pi) : \Pi \to \Re^n$ is a payoff table of utilities for each joint policy $\pi$ played by all players. $u^i(\pi)$ denotes the utility value for player $i$ and joint policy $\pi$. A player can choose a policy $\pi^i$ from $\Pi^i$ or sample from a mixture (meta-strategy) over them $\sigma^i \in \Delta\left(\Pi^i\right)$ where $\Delta$ is a probability simplex. In the terminology of game theory, $\sigma^i$ is a mixed strategy and each policy $\pi^i$ is a pure strategy.

Each player in the game is assumed to maximize their utility. The most well-known steady-state concept of a game is the Nash equilibrium (NE). NE is a strategy profile $\pi$ such that no player has an incentive to deviate from its current strategy given the strategies of the other players. Namely, each player's strategy is a best response to others' $\mathcal{BR}(\pi^{-i}) = \arg\max_{\pi^i} u^i(\pi^i, \pi^{-i}) \ \forall i \in \mathcal{N}$. We call a set of policies $\epsilon$-best responses to a joint opponents' policy $\pi^{-i}$, when there exists an $\epsilon > 0$, such that $\mathcal{BR}_\epsilon(\pi^{-i}) = \{\pi^i : u^i(\pi^i, \pi^{-i}) \geq u^i(\mathcal{BR}(\pi^{-i}), \pi^{-i}) - \epsilon\}$. An $\epsilon$-Nash equilibrium is a strategy profile that satisfies: $u^i(\pi) \geq \max_{\pi^{i\prime}} u^i(\pi^{i\prime}, \pi^{-i}) - \epsilon \ \forall i \in \mathcal{N}$.

## 3.2 Exploitability and Exploitation

To evaluate our learned policy $\pi^1$, we use two metrics. An agent's policy's $\pi^1$ exploitation of an opponent's policy $\pi^2$ is the extra gain obtained by the agent compared to its NE value $v^1$:

$$\omega(\pi^1, \pi^2) = u^1(\pi^1, \pi^2) - v^1.$$

This measures how much the policy $\pi^1$ exploits the weakness of the opponent's policy $\pi^2$. However, in general, there is no guarantee that the learned policy $\pi^1$ has no weakness. Therefore, we also define the exploitability of a policy $\pi^1$ which measures the loss incurred when the agent faces the

---

[i]A detailed definition of the trajectory in a stochastic game [38] can be found in Appendix A.1.

best opponent policy $\pi^2 = \mathcal{BR}(\pi^1)$ compared to the agent's Nash equilibrium value $v^1$:

$$
\begin{aligned}
\epsilon(\pi^1) &= v^1 - u^1\big(\pi^1, \mathcal{BR}(\pi^1)\big) \\
&= \max_{\pi^{1\prime}} \min_{\pi^2} u^1(\pi^{1\prime}, \pi^2) - \min_{\pi^2} u^1(\pi^1, \pi^2).
\end{aligned}
\tag{1}
$$

From Equation 1 we can see that the exploitability of a policy is non-negative and represents the distance of policy $\pi^1$ to an equilibrium.

### 3.3 Restricted Nash Response (RNR)

Johanson et al. [19] consider a modified zero-sum game where an opponent has a restricted strategy space $\Pi^2_{p,\pi_{\text{fix}}}$ such that it plays a fixed policy $\pi_{\text{fix}}$ with probability $p$ and plays any possible policy from the original strategy space $\Pi^2$ with probability $1 - p$. Given $(p, \pi_{\text{fix}})$, they define a restricted Nash equilibrium as a strategy profile $(\pi^{1*}, \pi^{2*})$ such that $\pi^{1*} \in \mathcal{BR}(\pi^{2*})$ and $\pi^{2*} \in \mathcal{BR}_{p,\pi_{\text{fix}}}(\pi^{1*})$, where: $\mathcal{BR}_{p,\pi_{\text{fix}}}(\pi^{1*}) = \arg\max_{\pi^2 \in \Pi^2_{p,\pi_{\text{fix}}}} u^2(\pi^1, \pi^2)$. It is shown that $\pi^{1*}$ is the best response to $\pi_{\text{fix}}$ among strategies which have equal or lower exploitabilities than $\pi^{1*}$, i.e.: $\pi^{1*} = \mathcal{BR}_{\epsilon}(\pi_{\text{fix}})$, where $\epsilon = \epsilon(\pi^{1*})$. Therefore, $\pi^{1*}$ is called a p-restricted Nash response (RNR) to $\pi_{\text{fix}}$. An RNR can be computed by solving the modified game, we present a linear programming solver implementation for NFGs in Appendix A.2.

## 4 Dirichlet Process Mixture Opponent Modeling

This section presents our non-parametric Bayesian method for modeling a non-stationary opponent. We consider an opponent's learning process as consecutive transitions from one policy to another such that one opponent can theoretically adopt infinitely many policies during its life-time. Therefore, we propose to use a Dirichlet process (DP) mixture to model the learning process as it has the ability to model an infinite number of clusters (policies in this case) while inferring the current number of policies from the data collected thus far. As our agent interacts with the opponent online, we learn a model with a sequential maximum-a-posteriori approach.

We model an opponent policy as a parameterized function $\pi^2_\phi$ and denote the parameter space as $\Phi$. To avoid cluttered notation in this section, we use $\phi$ to represent the modeled opponent's policy. $\mathbf{DP}(\alpha H)$ is a stochastic process with a concentration parameter $\alpha$ and a base distribution $H$ over $\Phi$. A random draw $G \sim \mathbf{DP}(\alpha H)$ is itself a distribution over $\Phi$, satisfying:

$$
(G(A_1), ..., G(A_r)) \sim \mathbf{Dirichlet}\big(\alpha H(A_1), ..., \alpha H(A_r)\big)
$$

for every finite measurable partition $A_1, ..., A_r$ of $\Phi$. The full graphical model for opponent modeling is shown in Figure 1a. It illustrates a generative process where at step $m$, the opponent first samples a policy $\hat{\phi}_m \sim G$ and then rolls-out this policy to collect a trajectory $\tau_m$.

To facilitate Bayesian inference, two representations of DP are considered. The stick-breaking representation in Figure 1b reveals the discrete nature of $G$. $G \sim \mathbf{DP}(\alpha H)$ can be constructed as $G = \sum_{k=1}^{\infty} \beta_k \delta_{\phi_k}$ where $\boldsymbol{\beta} \sim \mathbf{GEM}(\alpha)$ is an infinite-dimensional random variable sampled from the Griffiths-Engen-McCloskey (GEM) distribution and $\{\phi_k\}_{k=1}^{\infty}$ are i.i.d. sampled policies from $H$. At step $m$, the opponent samples a policy index $z_m \sim \mathbf{Categorical}(\boldsymbol{\beta})$ and rolls-out the policy $\phi_{z_m}$. Inference with the stick-breaking representation is required in order to handle the infinite dimensional $\boldsymbol{\beta}$. Therefore, the truncation method [6] is commonly used to limit the model capacity to a $K$ mixture and infer the actual number of policies by collapsing redundant ones. This requires tracking all $K$ policies simultaneously and does not adapt well to online settings.

The Chinese restaurant process (CRP) representation in Figure 1c can be obtained by integrating out $\boldsymbol{\beta}$. This introduces temporal dependencies between the policies, which can be expressed by the conditional distribution:

$$
p\big(z_{m+1} = k | z_{1:m}\big) = \begin{cases} \dfrac{\alpha}{m + \alpha}, & k = K_m + 1 \\[2ex] \dfrac{|k|_m}{m + \alpha}, & 1 \le k \le K_m \end{cases}
\tag{2}
$$

where $|k|_m = \sum_{i=1}^m \mathcal{I}(z_i = k)$ is the total number of trajectories from the $k$-th policy and $K_m$ is the number of realized policies up until step $m$. Inference with the CRP representation does not need to handle the infinite dimensional $\boldsymbol{\beta}$. Furthermore, at step $m$, we only need to track at most $m$ policies ($K_m \leq m$) while all policies beyond $K_m$ are independent from the collected trajectories $\tau_{1:m}$, and thus can be discarded from the model. In addition, the temporal dependencies between policies introduced by CRP can be used to develop an online learning algorithm.

Given sampled trajectories $\tau_{1:m}$, the target of our opponent model is to assign each trajectory to a policy and update existing policies with assigned trajectories. This can be achieved by seeking maximum-a-posteriori (MAP) estimations of $z_{1:m}$ and $\boldsymbol{\phi}_{1:K_m}$. To deal with streaming trajectories, the MAP algorithm should operate in an online fashion. Therefore, given the CRP representation, we decompose the posterior into the product of the posterior from the last step, the current priors and the likelihood, which leads to a recursive form:

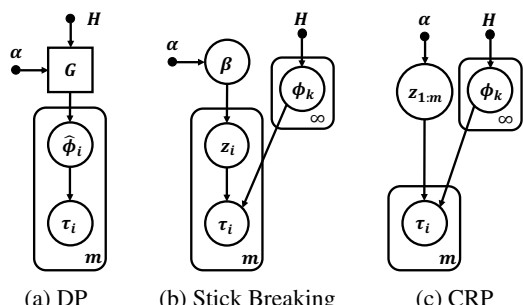

(a) DP      (b) Stick Breaking      (c) CRP

Figure 1: Dirichlet process mixture model

$$
\begin{aligned}
p(z_{1:m}, \boldsymbol{\phi}_{1:K_m}|\tau_{1:m}) &\propto \bigg( \prod_{k=1}^{K_{m-1}+1} p(\boldsymbol{\phi}_k) \bigg) p(z_1)p(\tau_1|\boldsymbol{\phi}_{z_1}) \bigg( \prod_{i=2}^m p(z_i|z_{1:i-1})p(\tau_i|\boldsymbol{\phi}_{z_i}) \bigg), \\
&\propto p(z_{1:m-1}, \boldsymbol{\phi}_{1:K_{m-1}}|\tau_{1:m-1})p(\boldsymbol{\phi}_{K_m})p(z_m|z_{1:m-1})p(\tau_m|\boldsymbol{\phi}_{z_m}),
\end{aligned}
$$
(3)

where $p(\boldsymbol{\phi}_k) = H$ is the base distribution of the DP.

The opponent model either assigns the current trajectory $\tau_m$ to a previous policy $\boldsymbol{\phi}_k$ or creates a new policy $\boldsymbol{\phi}_{K_{m-1}+1}$ to model $\tau_m$. The choice is made according to the MAP trajectory assignment $z_m^* = \arg\max_{z_m} p(z_m, z_{1:m-1}^*|\tau_{1:m})$:

$$
p(z_m = k, z_{1:m-1}^*|\tau_{1:m}) \propto
\begin{cases}
\int_{\boldsymbol{\phi}_k} \alpha p(\boldsymbol{\phi}_k)p(\tau_m|\boldsymbol{\phi}_k)\, d\boldsymbol{\phi}_k, \ k = K_{m-1}+1 \\
p(z_m = k|z_{1:m-1}^*)p(\tau_m|\boldsymbol{\phi}_k^{m-1}) = |k|_{m-1}^* p(\tau_m|\boldsymbol{\phi}_k^{m-1}), \ \text{otherwise}
\end{cases}
$$
(4)

where $|k|_{m-1}^* = \sum_{i=1}^{m-1} \mathcal{I}(z_i^* = k)$. Here, the hard assignment $z_m^*$ for $\tau_m$ is based on previous assignments $z_{1:m-1}^*$ and policies $\boldsymbol{\phi}_k^{m-1}$, which is equivalent to applying assumed density filtering (ADF) [41] to approximate the true posterior in Eq. (3) with a Delta distribution $\delta(z_{1:m}^*)$. The hard assignment prevents creating a new policy at each step if $\tau_m$ is assigned to an existing policy, which significantly reduces the memory usage. Furthermore, the MAP estimations for all existing policies, except $\boldsymbol{\phi}_{z_m^*}$, remain unchanged, which dramatically accelerates the algorithm. We then optimize the policy $\boldsymbol{\phi}_{z_m^*}$ by maximizing the likelihood of all trajectories assigned to it:

$$
\boldsymbol{\phi}_{z_m^*}^m = \arg\max_{\boldsymbol{\phi}_{z_m^*}} \log p(\boldsymbol{\phi}_{z_m^*}) \prod_{z_i^* = z_m^*} p(\tau_i|\boldsymbol{\phi}_{z_m^*}).
$$
(5)

Where finding the global optimum is not tractable in non-conjugate cases, we take gradient steps to update $\boldsymbol{\phi}_{z_m^*}^n$ as

$$
\boldsymbol{\phi}_{z_m^*}^n = \boldsymbol{\phi}_{z_m^*}^{m-1} + \lambda \nabla_{\boldsymbol{\phi}_{z_m^*}} \log p(\boldsymbol{\phi}_{z_m^*}) \prod_{z_i^* = z_m^*} p(\tau_i|\boldsymbol{\phi}_{z_m^*}).
$$

The entire algorithm fits into the general expectation-maximization (EM) framework. See Appendix A.4 for a detailed derivation.

The original CRP in Eq. (2) encapsulates a prior that the distribution of the next policy mimics the empirical policy distribution from the history. This prior is not consistent with our knowledge of the policy evolution process since a new opponent policy is commonly updated from the previous one. Therefore, we adopt a sticky variant in Eq. (6) to incorporate the belief that the opponent tends to

persist in the latest policy [11, 47].

$$p\big(z_m = k\big|z_{1:m-1}\big) = \begin{cases} \dfrac{\alpha}{m-1+\alpha+\kappa}, & k = K_{m-1}+1 \\[3mm] \dfrac{|k|_{m-1} + \kappa\hat{\delta}(K_{m-1},k)}{m-1+\alpha+\kappa}, & 1 \leq k \leq K_{m-1} \end{cases} \tag{6}$$

where $\kappa \geq 0$ is a 'stickiness' parameter and $\hat{\delta}$ is the Kronecker delta function.

Following Eq. (4), the probability of creating a new policy for $\tau_n$ is given by:

$$p(z_m^* = k) \propto \int_{\phi_k} \alpha p(\phi_k) p(\tau_m|\phi_k) \, d\phi_k, \tag{7}$$

where $k = K_{m-1}+1$. We use a Monte Carlo method to estimate Eq. (7) by sampling new policies from $p(\phi_k)$. However, sampling new policies from a data-independent prior $p(\phi_k)$ is likely to yield a low trajectory likelihood $p(\tau_m|\phi_k)$, which prevents the new policy creation. Therefore, we update the sampled policies to increase the likelihood $p(\tau_m|\phi_k)$ by taking a few gradient steps before estimating the integration in Eq. (7).

According to Eq. (6), the CRP prior encourages the opponent model to create redundant policies at the early stage when the number of trajectories $n$ is small and $\alpha$ dominates. Redundant policies could hurt the algorithm's performance as it incurs extra cost in terms of computation and memory. Trajectories from the same ground truth policy could be assigned to different $\phi_k$s and these assignments never revisited. Therefore, an error correction mechanism has to be introduced. Here, we adopt a symmetric distance metric between two policies and develop a policy merge procedure based on the metric. Given a set of states $\mathcal{S}$, we define $d(\phi_k, \phi_j) = \mathbb{E}_{s\sim\mathbf{Uniform}(\mathcal{S})}\left[\mathcal{JS}\Big(\phi_k(\cdot|s)\big\|\phi_j(\cdot|s)\Big)\right]$, where $\mathcal{JS}(\cdot\|\cdot)$ is the Jensen–Shannon divergence and $\phi_k(\cdot|s)$ is the action distribution given state $s$ under the policy $\phi_k$. When the distance between two policies is below a pre-defined threshold $\eta$, the merge procedure simply re-assigns all trajectories of $\phi_k$ to $\phi_j$.

With the opponent model developed in this section, at step $m$, we can construct an opponent policy set $\tilde{\Pi}^2 = \{\phi_k^m\}_{k=1}^{K_m}$ and a distribution $\tilde{\sigma}^2$ over $\tilde{\Pi}^2$. The distribution $\tilde{\sigma}^2(\phi = \phi_k) \propto |k|_m + \kappa\hat{\delta}(K_m,k)$ is essentially the empirical distribution of $z_{1:m}^*$ altered by the stickiness factor $\kappa$.

## 5 Exploit Policy-Space Opponent Model

In this section, we present how to learn a safe best response to this meta strategy, given a predicted distribution $\tilde{\sigma}_\phi^2$ [ii] over opponent's policies. The advantages of our approach of focusing on policy space are two-fold: first, we do not need to assume the access to the opponent's learning characteristics such as its training algorithm, its neural network's architecture or its update frequency; we only require past trajectories. Additionally, the distribution of the types of opponent policy $\tilde{\sigma}^2$ gives us an approximate stable overview of the current opponent's playing behavior compared to the opponent's current policy whose updates greatly depend on the opponent's learning characteristics and randomness from playing (e.g. exploration behavior) and training (e.g. stochastic gradient descent). Therefore, learning a response to this meta-strategy $\tilde{\sigma}^2$ will rely on less prior knowledge about the opponent's learning characteristics and is more robust to noise.

However, there is no guarantee that our learned meta strategy $\sigma^1$ has no (or at least low) exploitability. It has been shown that overfitting to an opponent strategy $\tilde{\sigma}^2$ often renders the resulting learned strategy brittle [19, 21, 46]. Such a brittle strategy performs badly when playing against different opponent strategies $\tilde{\sigma}^{2\prime}$. Therefore, a more desirable goal is to learn a safe best response to an opponent meta-strategy $\tilde{\sigma}^2$. RNR solutions consider cases where the game to solve is fixed and known and the opponent's policy is stationary. However, when we consider non-stationary opponent exploitation on policy space, the size of the meta-game to solve increases with the number of interactions between the training agent and the adaptive opponent. Furthermore, each player is free to learn and update its policy at any time point during the process.

---

[ii]To simplify our notation, we will ignore the subscript $\phi$ henceforth.

**Algorithm 1:** Exploit Policy-Space Opponent Model (EPSOM)

---

**input** : Hyper-parameter $p, H, E$, an adaptive opponent $-i$
**output** : Policy $\pi^i_{1,\ldots,E}$ and meta-policy $\sigma^i$
Initialize learning agent $i$'s policy $\pi^i_0$
Initialize a memory buffer **B**
Initialize opponent meta-policy $\sigma^{-i}(\cdot) = 1$
**for** *epoch e in* $\{1, 2, \ldots, E\}$ **do**
    **for** *episode* $h \in \{1, 2, \ldots, H\}$ **do**
        Play an episode against the opponent with strategy $\sigma^1_{RNR}$
        Collect the trajectory $\tau_{e,h}$ and save them into **B**
    **end**
    $\tilde{\sigma}^2, \tilde{\Pi}^2 = \text{opponent\_modeling}(\mathbf{B})$
    $\bar{p} = \frac{1}{|\tilde{\Pi}^2|} \sum_j p^j \tilde{\sigma}^2(j)$
    Compute missing entries in $U^{\tilde{\Pi}}$ from $\tilde{\Pi} = \Pi^1 \times \tilde{\Pi}^2$ by simulations
    $\_, \sigma^2_{RNR} = \text{RNR\_solver}(U^{\tilde{\Pi}}, \bar{p}, \tilde{\sigma}^2)$
    **for** *episode* $h \in \{1, 2, \ldots, H\}$ **do**
        Sample $\tilde{\pi}^2$ from $\sigma^2_{RNR}$
        Train oracle $\pi^1$ over $\rho \sim (\pi^1, \tilde{\pi}^2)$
    **end**
    $\Pi^1 = \Pi^1 \cup \{\pi^1\}$
    Compute missing entries in $U^{\tilde{\Pi}}$ from $\tilde{\Pi} = \Pi^1 \times \tilde{\Pi}^2$ by simulations
    $\sigma^1_{RNR}, \_ = \text{RNR\_solver}(U^{\tilde{\Pi}}, \bar{p}, \tilde{\sigma}^2)$
**end**

---

To address the above issues, we combine DO with RNR to solve a meta-game built from EGTA where the opponent's policies are predicted by the opponent model from Section 4. Pseudo-code explaining our approach is presented in Algorithm 1. We maintain a utility table $U^{\tilde{\Pi}}$ wherein rows represent learned policies for the training agent and columns represent a modeled policy of the opponent respectively. An epoch is defined as a fixed amount of episodes of games where we play against the opponent holding our strategy $\sigma^1$ fixed. At each epoch, we run our opponent model to predict the current distribution $\tilde{\sigma}^2$ of the opponent's policies. If a new policy is detected, we will add it into $\tilde{\Pi}^2$. Given $\tilde{\sigma}^2$, we run a p-RNR solver to obtain the opponent's RNR meta-strategies $\sigma^2_{RNR}$ which is a restricted Nash solution to the current meta-game assuming that the opponent is playing according to $\tilde{\sigma}^2$ with probability at least $p$. Then we train an (approximate) best-response policy to $\sigma^2_{RNR}$ and add the new policy into $\Pi^1$. We re-run a p-RNR solver to obtain our RNR meta-strategies $\sigma^1_{RNR}$ which we use to mix the policies in population $\Pi^1$ for the next epoch's playing policy.

When a new type $j$ of modeled policy $\pi^{2,j}$ is added by our opponent model, we initialize a p-value $p^j = p_{init}$ to this type. Its p-value is incremented proportionally to the probability that the opponent plays this policy in the following epochs, $\tilde{\sigma}^2(j)$, and clipped at 1. At each epoch, we calculate the average p-value $\bar{p} = \frac{1}{|\tilde{\Pi}^2|} \sum_j p^j \tilde{\sigma}^2(j)$ for solving current RNR strategies. In the extreme case where $\bar{p} = 0$, $\sigma^i_{RNR}$ is the same as the Nash strategy in the current meta game. At another extreme where $\bar{p} = 1$, $\sigma^2_{RNR} = \tilde{\sigma}^2$ and $\sigma^1_{RNR} = \mathcal{BR}(\tilde{\sigma}^2)$. Therefore, when we have low confidence in $\tilde{\sigma}^2$ ($\bar{p}$ is low), we learn an approximate best response to opponent's current Nash mixture which will enlarge our current empirical gamescape [3] and thus help to find strategies with lower exploitability. At the same time, the training agent becomes risk-adverse and the next epoch strategy $\sigma^1_{RNR}$ becomes a strategy closer to the Nash strategy of the current meta game.

When $\bar{p}$ is high, it means that our opponent model has high confidence that the opponent is playing $\tilde{\sigma}^2$ and an approximate best counter strategy to $\tilde{\sigma}^2$ will be added into our policy population. The training agent becomes profit-driven and $\sigma^1_{RNR}$ becomes a strategy closer to $\mathcal{BR}(\tilde{\sigma}^2)$ in the next epoch. Therefore, mixing the playing policy by $\sigma^1_{RNR}$ flexibly switches the agent between risk-adverse and profit-driven depending on the confidence of the opponent model. In contrast with previous RNR solution, EPSOM can always recover a strategy with approximately the lowest exploitability it has seen so far as we maintain a population of policies.

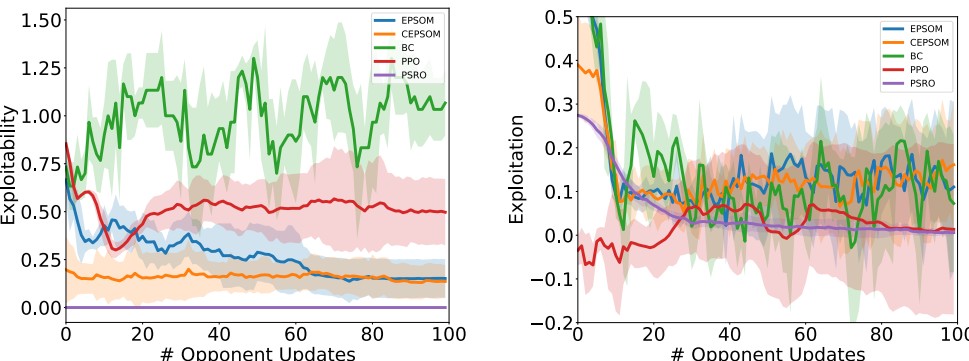

Figure 2: Exploitability and exploitation of different algorithms against a non-stationary opponent implemented by PPO in Kuhn Poker.

## 6 Experiments

In this section, we empirically investigate whether the proposed method can (1) exploit an unknown non-stationary opponent while still maintaining a strategy with low exploitability, (2) improve its performance by continued training against different opponents and (3) exploit previously unseen opponents without further training. We verify EPSOM's performance in Kuhn Poker [20], a simplified version of poker which importantly retains strategic elements useful for game-theoretic analysis. We use an agent learning to play using the PPO [37] algorithm as our opponent, which suffices to provide a non-stationary setting. More details about the game and our experiment implementation can be found in Appendix A.3.

| Kuhn Poker | | | | |
|---|---|---|---|---|
| | PPO | | TRPO | |
| EPSOM [0.109] | 0.023 | (0.189) | 0.09 | (0.180) |
| CEPSOM [0.080] | **0.113** | (0.142) | **0.140** | (0.117) |
| BC [1.333] | $-0.559$ | (0.232) | $-0.223$ | (0.134) |
| PSRO [0.000] | 0.052 | (0.072) | 0.032 | (0.058) |
| PPO [0.477] | $-0.407$ | (0.133) | $-0.372$ | (0.119) |

Table 1: Zero-shot learning exploitation results. Trained agents (row players) play against adaptive opponents (column players) without further training. Adaptive opponents are allowed to update 100 times and a trained agent's average exploitation are taken over these 100 updates and 5 random seeds. Values in square brackets are each trained agent's exploitability and values in parentheses are stds taken over 5 random seeds.

Figure 3: Opponent's learning process modeled by trained CEPSOM: (left) CEPSOM adjusts playing strategy online; (right) CEPSOM plays an approximate Nash strategy. Each point is 2-D embedding of a modeled policy to which a trajectory produced by the opponent is assigned by our opponent model. Color bar indicates time sequence.

We select 3 representative algorithms as baselines. PSRO is a popular algorithm which guarantees the convergence to an approximate NE. As PSRO is a self-play algorithm, its is trained before playing against any adaptive opponents. Behavior cloning (BC) models the opponent's policy by taking maximum likelihood estimation of history trajectories stored in a sliding-window buffer and learns an (approximate) best-response policy to it. PPO represents a canonical choice among many SARL algorithms. In our work, as agents and their opponents update asynchronously, we always evaluate each algorithm's performance right after the opponent's update for a more robust evaluation. The following results reported with mean and standard deviation (std) are all obtained by repeating the corresponding experiment over 5 random seeds.

As shown in Figure 2, EPSOM can achieve a safe strategy with relatively low exploitability while still being able to exploit its opponent. Though PSRO plays a strategy with the lowest exploitability ($\approx 0$) it also has very low exploitation against its opponent. In contrast, BC can exploit its opponent to a similar extent as EPSOM but it comes with the cost of high exploitability. The PPO algorithm has

large variance and performs badly on average in this non-stationary setting. We also test a continual learning version of EPSOM which we name CEPSOM. It is implemented by training an EPSOM agent against 5 different opponents without re-initialization thereby building up a richer set of modeled opponent policies and a more robust best-response policy population. Its average performance over these opponents is also reported in Figure 2. In our experiments, we use an analytical method to calculate a best response to a given policy. We obtain similar results for approximate best response learned by RL which are reported in Appendix A.5.

Next, we test these agent's performance against two adaptive opponents implemented by PPO and TRPO [36] without further training and results are presented in Table 1. Relying on an opponent model to predict the current opponent's policy type and flexibly adjusting its playing strategy accordingly, CEPSOM achieves the highest average exploitation against adaptive opponents. EPSOM also obtains positive average exploitation but with a much lower value, since EPSOM has only ever been trained with one opponent. PSRO plays a safe strategy and performs slightly better than EPSOM in terms of opponent exploitation. BC and PPO perform badly as they overfit to one opponent, and thus they are exploited by other adaptive opponents. Note that, in this zero-shot learning tournament, although we do not train EPSOM and CEPSOM, they still need to predict the opponent's policy and solve the meta-game for a RNR solution given the prediction.

Facing an adaptive opponent, a trained CEPSOM's opponent model can assign trajectories collected during learning into modeled policies it has built. Therefore, we can visualize an opponent's learning process by presenting a sequence of those modeled policies on a 2-dimensional plot. In Figure 3, we present two opponent's visualized learning process when it faces (a) a trained CEPSOM agent which adjusts its playing strategy online based on its opponent model prediction, and (b) a trained CEPSOM agent which always plays an approximate Nash equilibrium strategy. We can see that in a non-stationary environment (left), the opponent's learning exhibits a cyclic pattern. In contrast, the opponent's learning is much more transitive (i.e., monotonically moving in one direction in the 2D space) in a stationary environment (right).

# 7 Conclusion

In this work, we propose a framework for training an agent to safely exploit its opponent. Compared to RNR and its variants, our work focuses on non-stationary opponents. We consider the opponent's learning as a series of policy transitions and model such a process by a Dirichlet Process. Safe exploitation means that an agent can exploit an opponent's weakness to maximize our utility while simultaneously maintaining a strategy which has low exploitability. This property is desirable as naively overfitting to one type of opponent could easily lead to exploitation by other opponents. We empirically verify our algorithm's performance on Kuhn Poker, a simplified version of Poker.

Opponent modeling based MARL algorithms typically require extra computation for learning a good opponent model. This cost often scales dramatically with the number of opponents, action space dimensionality and the complexity of the problem. It can be a heavy burden on an agent if it learns an opponent model from scratch online. Therefore, a more realistic way for utilizing the power of an opponent model is offline training and online prediction. We build CEPSOM based on this idea where we train one EPSOM agent across different opponents and aggregate knowledge by maintaining a never-reinitialized opponent model and policy population. Our experiment results show that CEPSOM can achieve high exploitation against a new adaptive opponent without further training, outperforming other representative baselines from SARL and MARL. In complex competitive games, a strong player can often encounter sub-optimal opponents and playing a Nash strategy can potentially forego significant profit. EPSOM, alongside many prior works, shows the potential of an opponent modeling based approach for solving this problem, and our preliminary results from CEPSOM demonstrate the possibility of a trained agent beating an as yet unseen adaptive opponent.

EPSOM is limited by its computation and memory complexity. Naively applying EPSOM to more complex problems requires a great amount of resources. To alleviate this problem, we introduce policy merge to remove redundant policies in our opponent model. This approach could be improved by applying game theoretic analysis to our policy populations (agent's self policies and modeled opponent policies). We leave the study of improving EPSOM's scalability to future work.

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
