# A  Appendix

## A.1  Stochastic Game

We assume that interactions of players in an environment can be modeled by a stochastic game [38]. For a 2-agent stochastic game, we define a tuple $(\mathcal{S}, \mathcal{A}^1, \mathcal{A}^2, r^1, r^2, p, \gamma, \mathcal{N})$, where $\mathcal{N} = \{1, 2\}$, $\mathcal{S}$ is the state space, $p$ is the distribution of the initial state $s_0$, $\gamma \in [0, 1]$ is the discount factor for future rewards, $\mathcal{A}^i$ and $r^i = r^i(s, a^i, a^{-i})$[iii] are the action space and the reward function respectively for agent $i \in \{1, 2\}$. Agent $i$ chooses its action $a^i \in \mathcal{A}^i$ according to the policy $\pi^i(a^i|s)$ conditioning on the state $s \in \mathcal{S}$.

Where the environment is partially observable, an agent's observation at time $t$ is defined as $o_t^i$ which is generated from the state $s_t$ by an unknown observation function $F(s, i) : \mathcal{S} \times \mathcal{N} \to \mathcal{O}^i$ where $\mathcal{O}^i$ is agent $i$'s observation space. To resolve issues resulting from partial observability, we use the history of a game $h_t^i = \{o_{1:t}^i, a_{1:t-1}^i, a_{1:t-1}^{-i}\}$ so far from time step $t$ as the input to the policy $\pi^i(a^i|h^i)$. In this work, we specifically denote a trajectory as a set of the opponent's end-game histories collected from $d$ episodes $\tau = \{h_{1,T}^2, h_{2,T}^2, \dots, h_{d,T}^2\}$, where $T$ is the terminal time step in an episode. This implies that the opponent's policy does not change over $d$ episodes. However, this is not true in our setting or in general. Therefore, we take $d$ as a hyper-parameter and tune it to obtain satisfactory empirical results.

## A.2  RNR Solver

An RNR solution for a normal-form game considers a modified game where the opponent plays a fixed strategy $\pi_{\text{fix}}$ with probability $p$ and any strategy from its original strategy space $\Pi^2$ with probability $1 - p$. We implement a linear programming solver for RNR and present our formulation as follows:

We use $\pi^1$ and $\pi^2$ to denote an agent's and its opponent's strategies respectively. Further, let $U^\Pi$ denote the utility table for the agent. Since we consider the zero-sum game, the utility table for the opponent is simply $\text{trans}(-U^\Pi)$ where $\text{trans}$ denotes the matrix transpose operation.

As the opponent is restricted to play a fixed strategy $\pi_{\text{fix}}$ with with probability at least $p$, the overall opponent policy satisfies:

$$p\pi_{\text{fix}}(a) \leq \pi^2(a) \leq p\pi_{\text{fix}}(a) + 1 - p \quad \forall\, a \in \Pi^2.$$

We define a vector $y_a$ of size $|\Pi|^2 \times 1$ whose element indices correspond to opponent actions, and we have:

$$y_a(j) = \begin{cases} 0, & j \neq a \\ p\pi_{\text{fix}}(a) + 1 - p, & j = a. \end{cases}$$

Therefore, for $\pi_{RNR}^1$, we solve the following linear programming problem:

$$\max u$$
$$\text{s.t. } u \leq {\pi^1}^T U^\Pi y_a \quad \forall\, a \in \Pi^2,$$
$$\pi^1(b) \geq 0 \quad \forall\, b \in \Pi^1,$$
$$\sum_{\Pi^1} \pi^1(b) = 1.$$

Similarly, for $\pi_{RNR}^2$, we solve the following linear programming problem:

$$\min v$$
$$\text{s.t. } v \geq x_b^T U^\Pi \pi^2 \quad \forall\, b \in \Pi^1,$$
$$\pi^2(a) \leq p\pi_{\text{fix}}(a) + 1 - p \quad \forall\, a \in \Pi^2,$$
$$\pi^2(a) \geq p\,\pi_{\text{fix}}(a) \quad \forall\, a \in \Pi^2,$$
$$\sum_{\Pi^2} \pi^2(a) = 1,$$

---

[iii]We use subscript $-i$ to denote the complementary part $x^{-i}$ of the variable $x^i$ indexed by $i$.

where we define a vector $x_b$ of size $|\Pi^1| \times 1$ whose each element index corresponds to an agent's action, and we have:

$$x_b(j) = \begin{cases} 0 & j \neq b \\ 1 & j = b. \end{cases}$$

### A.3 Experiment Details

Variants of Poker offer a rich arena for developing artificial intelligence. The games feature stochasticity, partial observability and competitive dynamics with unknown adversaries. In this work, we conduct experiments and evaluation in a simplified Poker game, Kuhn Poker. This variant of poker is amenable to game theoretic analysis whilst retaining all of the elements of the more challenging larger scale poker games. In Poker, the players take it in turns to bet with the knowledge that should betting conclude with both players still in the game, the player with the highest scoring hand wins. In this section, we explain the rules of Kuhn Poker.

**Kuhn Poker [20]**] is a simple, zero-sum two-player imperfect information game. The deck of cards is limited to simply a Jack, a Queen and a King with no notion of suits. Ordering is as usual: Jack < Queen < King. If the game reaches a showdown, the player with the highest card wins. If either player folds at any time they lose the round and their opponent takes the entire pot. The game opens with a round of antes of 1. Then each player is dealt a single card and the remaining card is placed face down. Once the deal is complete it is time for the first round of bidding: Player 1 may check (no bet) or bet 1. If Player 1 bet Player 2 may call the bet or fold. If Player 2 calls there is a showdown for the pot of 4, if they fold Player 1 wins the pot. If Player 1 checked, Player 2 may check or bet 1. If both players check then there is a showdown for the pot of 2. If Player 2 bets, following a check by Player 1, then Player 1 can either fold or call. If player 1 calls there is then a showdown for the pot of 4. The starting player may alternate or be chosen at random for each deal.

Kuhn Poker has the second-mover advantage, i.e., the second player to bet (Player 2 above) will win in expectation when both players play the best response to each other. To remove this advantage, we alternate the playing turn between our agent and the opponent after every episode of a game. In this simple game, we do not discriminate between the Pass and Fold actions, and thus, each player need only choose from Pass or Bet. Our hyperparameter values are presented in Table 2.

Table 2: Hyper-parameter settings.

| SETTINGS | VALUE | DESCRIPTION |
|---|---|---|
| **EPSOM** | | |
| ORACLE METHOD | ANALYTICAL BEST RESPONSE | SUBROUTINE OF GETTING ORACLES |
| $d$ | EVERY 64 EPISODES OF A GAME | UPDATE FREQUENCY FOR EPSOM |
| META-SOLVER | LINEAR PROGRAMMING SOLVER | META-SOLVER METHOD |
| **DIRICHLET PROCESS MIXTURE OPPONENT MODEL** | | |
| $p_{init}$ | 0.1 | INITIAL $p$ VALUE FOR A NEW POLICY |
| $p_{step}$ | 0.05 | $p$ VALUE INCREMENT WHEN A NEW TRAJECTORY IS ASSIGNED TO THE POLICY |
| $\alpha$ | 1.0 | CONCENTRATION PARAMETER OF THE DIRICHLET PROCESS |
| $\kappa$ | 1.0 | 'STICKINESS' FACTOR FOR MODIFIED CRP PRIOR |
| $\theta$ | 5.0 | STD OF THE POLICY BASE DISTRIBUTION: $p(\phi_k) = \mathcal{N}(0, \theta^2 I)$ |
| $\eta$ | 0.1 | THRESHOLD FOR MERGING TWO POLICIES INTO ONE |
| **PPO OPPONENT** | | |
| LEARNING RATE | 0.0003 | LEARNING RATE FOR PPO |
| OPTIMIZER | ADAM | GRADIENT ASCENT OPTIMIZER |
| NN ARCHITECTURE | $12 \times 64 \times 64 \times 2$ | NEURAL NETWORK ARCHITECTURE |
| MINI BATCH SIZE | 128 | MINI BATCH SIZE FOR SGD |
| UPDATE FREQUENCY | EVERY 128 STEPS | OPPONENT UPDATE AFTER EVERY 128 STEPS |
| UPDATE EPOCH | 20 | TRAINING EPOCHS IN AN UPDATE |
| CLIP RATIO | 0.2 | PPO CLIP RATIO |
| $\gamma$ | 0.99 | DISCOUNT FACTOR |
| $\lambda$ | 0.97 | LAMBDA-RETURN FACTOR |
| **KUHN POKER** | | |
| OBSERVATION DIMENSION | 12 | 12 COMBINATIONS FOR SELF HAND AND GAME HISTORY |
| ACTION SPACE | 2 | PASS OR BET |

### A.4 Streaming MAP for Opponent Modeling

Our streaming MAP algorithm for opponent modeling fits into the general expectation-maximization (EM) framework. In the E-step, we seek a Delta distribution $q(z_{1:m}) = \delta(z_{1:m}^*)$ to approximate the posterior $p(z_{1:m}|\tau_{1:m})$. Therefore, $z_{1:m}^*$ are the MAP trajectory assignments. Following Eq. (3),

$p(z_{1:m}|\tau_{1:K_m})$ can be obtained by integrating out $\phi_{1:K_m}$:

$$p(z_{1:m}|\tau_{1:m}) = \int_{\phi_{1:K_m}} p(z_{1:m}, \phi_{1:K_m}|\tau_{1:m})\, d\phi_{1:K_m}$$

$$\propto \int_{\phi_{1:K_m}} p(z_{1:m-1}, \phi_{1:K_{m-1}}|\tau_{1:m-1}) p(\phi_{K_m}) p(z_m|z_{1:m-1}) p(\tau_m|\phi_{z_m})\, d\phi_{1:K_m}.$$

$$(8)$$

To solve the E-step in an online fashion, we apply assumed density filtering [40] and approximate $p(z_{1:m-1}, \phi_{1:K_{m-1}}|\tau_{1:m-1})$ in Eq.(8) with $q(z_{1:m-1})q(\phi_{1:K_{m-1}}) = \delta(z_{1:m-1}^*)\delta(\phi_{1:K_{m-1}}^{m-1})$. There-fore, $q(z_{1:m})$ is computed recursively by reusing $q(z_{1:m-1})$ and the policies $\phi_{1:K_{m-1}}$ are fixed to their latest value $\phi_{1:K_{m-1}}^{m-1}$. Since a new policy may be created, we also need to integrate over $\phi_{K_{m-1}+1}$ to incorporate the possibility that $K_m = K_{m-1} + 1$.

---

**Algorithm 2:** Streaming MAP for Opponent Modeling

---

**Initial Step:**
Solve $z_1^*, \phi_{z_1}^1 = \arg\max_{z_1^*, \phi_{z_1}} p(\phi_{z_1})p(z_1)p(\tau_1|\phi_{z_1})$.
**for** $m = \{2, 3, \dots\}$ **do**

    **Update $z_{1:m}^*$ (E-Step):**
    Maximum-a-posterior (MAP) trajectory assignments $z_{1:m}^*$:

$$z_{1:m}^* = \arg\max_{z_{1:m}} p(z_{1:m}|\tau_{1:m})$$

$$\approx \arg\max_{z_{1:m}} \int_{\phi_{1:K_{m-1}+1}} q(z_{1:m-1})q(\phi_{1:K_{m-1}}) p(\phi_{K_{m-1}+1}) p(z_m|z_{1:m-1}) p(\tau_m|\phi_{z_m}) d\phi_{1:K_{m-1}+1}$$

    $z_{1:m-1}^*$ remains unchanged and

$$z_m^* = \arg\max_{z_m} \int_{\phi_{1:K_{m-1}+1}} q(\phi_{1:K_{m-1}}) p(\phi_{K_{m-1}+1}) p(z_m|z_{1:m-1}^*) p(\tau_m|\phi_{z_m}) d\phi_{1:K_{m-1}+1}$$

    which corresponds to Eq. (4). Then $q(z_{1:m}) = \delta(z_{1:m}^*)$ and $K_m$ is set according to $z_m^*$.

    **Update $\phi_{1:K_m}^m$ (M-Step):**

$$\phi_{1:K_m}^m = \arg\max_{\phi_{1:K_m}} \mathbb{E}_{p(z_{1:m}|\tau_{1:m})} \left[ \log p(\tau_{1:m}, z_{1:m}|\phi_{1:K_m}) \right] + \log p(\phi_{1:K_m})$$

$$\approx \arg\max_{\phi_{1:K_m}} \int_{z_{1:m}} q(z_{1:m}) \log \prod_{i=1}^m p(\tau_i|\phi_{z_i}) dz_{1:m} + \log \prod_{k=1}^{K_m} p(\phi_k)$$

$$= \arg\max_{\phi_{1:K_m}} \log \prod_{k=1}^{K_m} p(\phi_k) \prod_{i=1}^m p(\tau_i|\phi_{z_i^*})$$

    For $k \neq z_m^*$, $\phi_k^m = \phi_k^{m-1}$ and $\phi_{z_m^*}^m = \arg\max_{\phi_{z_m^*}} \log p(\phi_{z_m^*}) \prod_{z_i^*=z_m^*} p(\tau_i|\phi_{z_m^*})$ which
    corresponds to Eq. (5).
**end**

---

In the M-step, we optimize the policies $\phi_{1:K_m}$ to maximize $\mathbb{E}_{q(z_{1:m})}\left[\log p(\tau_{1:m}, z_{1:m}|\phi_{1:K_m})\right]$ with an extra regularization term, $\log p(\phi_{1:K_m})$, introduced by the policy prior. The whole procedure is summarized in Algorithm 2. For the initial step, the trivial solution is given by: $z_1^* = 1, \phi_1^1 = \arg\max_{\phi_1} p(\phi_1)p(\tau_1|\phi_1)$.

### A.5 RL oracle

We test EPSOM with an RL oracle which is implemented by a PPO algorithm. In these experiments, we also use PPO as an RL oracle for our PSRO and BC baselines. The PPO oracle uses the same set

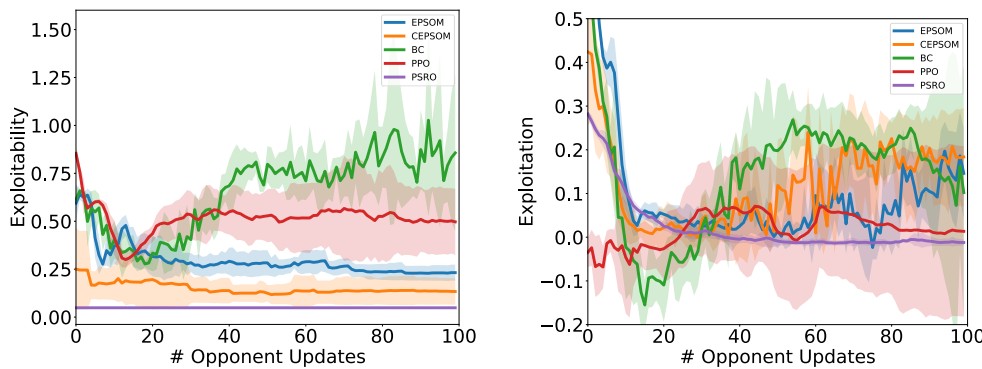

Figure 4: Exploitability and exploitation of different algorithms against a non-stationary opponent implemented by PPO in Kuhn Poker.

of hyper-parameters as the PPO implemented for the opponent and we present our results in Figure 4. We obtain similar results as we are obtained in the case of using analytical best responses.