# OpenReview forum: "Learning to Safely Exploit a Non-Stationary Opponent"
_NeurIPS.cc/2021/Conference — NeurIPS 2021 Submitted_

### Official Review · Reviewer_hxK1 · 2021-07-08

**Rating:** 6
**Confidence:** 4

**Summary:**

This work proposes an approach to learning against non-stationary opponents, where the non-stationarity comes from the learning dynamics in contrast to previous work which assumes a set of possible strategies over the opponents.

The opponent model is learned with Dirichlet Process and from it it's used together with a previous work (RNR) and meta learning  to produce a safe strategy.

The approach seems novel and improving previous works. I have some questions about experiments and limitations.


**Limitations And Societal Impact:**

No discussion on this topic

**Main Review:**

Originality
-I think a relevant work that is missed is "Safe Opponent Exploitation"
-I also would like to understand more about your contribution and previous work related to:
"Recently proposed streaming inference algorithms [23, 41] enable the DP mixture model to solve online non-stationary problems in a truly non-parametric way.  In this paper, we adopt this approach to model and simulate a non-stationary opponent for MARL."

Quality
- I would have liked to see ablation results for the "correction" mechanism, how robust are the result with respect to the hyperparameter $eta$
- The paper mentions many times that the approach learns online, however in the conclusions
"It can be a heavy burden on an agent if it learns an opponent model from scratch online. Therefore, a more realistic way for utilizing the power of an opponent model is offline training and online prediction"
 I'm a bit confused which parts are online, which parts offline.


Clarity
- "Our approach, EPSOM, is not limited to the self-play setting."
 I think EPSOM should *not* be used in self-play?
- Figure 1 needs a better a more descriptive caption
- I really like the exploitability and exploitation measures, I think you can add another result where those are the axes and you plot the results of different algorithms.
- Figure 3 is a bit hard to interpret
- Line 275: "the training agent becomes risk-adverse"  why?

Significance
- Would this approach work for general sum games?
- Would be interesting to see more experiments with more opponents and see how it scales
- Both PPO and TRPO are algorithms that limit the change in the policy from one step to another, I would be interested to see EPSOM with more "drastic" changes.
- "EPSOM is limited by its computation and memory complexity. Naively applying EPSOM to more
 complex problems requires a great amount of resources." I appreciate the comment, however, I think more details would improve the paper.

Minor:
Line 290: "use an agent learning"

**Time Spent Reviewing:**

4

---

> ### Author Response · Authors · 2021-08-10
> **Comments for Reviewer 4**
>
> Thank you for taking time to review our work and providing detailed and insightful feedback.
>
> Addressing your questions:
>
> 1. Safe Opponent Exploitation” (Ganzfried and Sandholm 2012): This paper is indeed a relevant work to us and we will include it in our updated version. Thank you for bringing it to our attention.
>
> 2. Discussion on work [23, 41]: Both [23] and [41] develop sequential variational inference algorithms for the Dirichlet Process Mixture Model (DPMM). [23] achieved this by introducing a heuristic approximation (Eq. 10 in [23]) to the DP posterior while [41] presented a theoretically justified framework leveraging assumed density filtering. Both papers reach the same updating rule for the policy indexes $z_{1:m}$ but differ from their updating rule for the policies $\phi_{1:K_m}$ (Eq.16 in [23] and Eq. 16 in [41]) . The policy update in [41] is biased towards the latest trajectory $\tau_m$ while previous trajectories $\tau_{1:m-1}$ are encoded as the prior. It can easily cause a practical problem that the prior can be weak and the likelihood of the latest trajectory dominates, especially when policies are updated by a stochastic gradient method. Our policy updating is similar to [23] where the likelihood of all trajectories $\tau_{1:m}$ assigned to the cluster are used. Instead of introducing a heuristic approximation like [23], we theoretically justify our algorithm by combining the assumed density filtering with the expectation-maximization framework (See appendix A.4). In contrast to previous works, we use a delta distribution to approximate the posterior of $z_{1:m}$, which is motivated by our problem setting (explained in section 4 line 200-204).
>
> 3. Value of eta: The value of eta essentially controls how easily we merge two modeled policies clusters into one. The smaller the value is, the less likely that two policies will be merged. When we set eta to 0.001, we do not observe merges during our training any more. When we rerun our experiments with eta values at 0.001, 0.01,  0.125, 0.15, 0.2 and 0.3. As we cannot present the relevant learning curves by figures as we did in Figure 2 here, we will summarise its learning curves by mean and stds over PPO opponent’s updates in this rebuttal. During its training against PPO opponent, CEPSOM can achieve exploitability at 0.167 (0.128), 0.095 (0.106), 0.249 (0.121),0.178 (0.116), 0.209 (0.137), 0.270 (0.181) respectively and exploitation of 0.145 (0.128), 0.110 (0.156), 0.135 (0.180), 0.143 (0.195), 0.121 (0.182), 0.113 (0.189).  Similarly, for EPSOM, it can achieve average exploitability of 0.257 (0.130), 0.250 (0.133), 0.259 (0.150), 0.290 (0.127), 0.276 (0.128), 0.346 (0.148) and exploitation at 0.197 (0.245), 0.124 (0.191), 0.126 (0.205), 0.167 (0.246), 0.147 (0.226), 0.098 (0.208) respectively. As we can see, large eta values will cause two policies which should be modelled separately being merged into one and this hurts the performance of EPSOM and CEPSOM. EPSOM is affected more by this because it only sees one opponent during its training. However, a small eta value will create unnecessary policies and make the computation of RNR solutions slow.
>
> 4. Online and Offline parts: We apologise for the poor clarity of our presentation on this point. The training of original EPSOM and CEPSOM (including the learning of the opponent model, response policies and solving for RNR solutions on the meta-level game) is online. However, as discussed in the paper, the computation cost is heavy, especially when we learn the opponent model. Therefore, we want to check, once trained and without any further training, how well EPSOM and CEPSOM will perform against new opponents which are able to learn and adapt. These results are summarised in Table 1 and we can see both EPSOM and CEPSOM can maintain positive exploitation against these opponents. Based on this observation, we then claim we could alleviate the heavy online computation cost problem by moving the learning of the opponent model and response policies offline. Namely, before we start a real tournament and play against real opponents, we can first train EPSOM and CEPSOM with ‘pseudo’ opponents created by us. Then in the real tournament, we do not need to train EPSOM and CEPSOM but only solve RNR solutions for the meta-game online based on the prediction of the opponent model which is trained offline (before the tournament).
>
> 5. Exploitability and exploitation plot: In fact, we first attempted to plot our figures in the way you suggest. However, the significant fluctuations of the baseline algorithms such as BC and PPO can make the whole plot difficult to read. Therefore, we changed the format to the current version.
> 6. “Risk-adverse” problem: This is a typo and we meant to say ‘risk-averse’. Our agent becomes risk-averse because it has low confidence in the prediction of the opponent model. As a result, it will play a strategy with low exploitability to minimise the potential risk.  Recall, whenever a cluster (a modelled policy) is revisited its p-value will be accumulated by a fixed amount (0.05 in our work). Therefore, the weighted average p-value can tell us how many times the predicted clusters have been revisited. This could be an indicator of confidence about the current prediction.
> 7. General-sum games: Our work cannot easily be extended to general-sum games.
> 8. Opponent with more “drastic” changes: We added A2C opponents as a new type in our updated version. Once trained, EPSOM, CEPSOM, BC, PSRO and PPO can achieve exploitation of 0.146 (0.044), 0.188 (0.137), -0.419(0.184), 0.064(0.094), -0.377(0.147) against A2C opponents respectively. As can be seen, CEPSOM is still the best algorithm which can exploit an adaptive A2C agent without any further training.

---

> ### Comment · Reviewer_hxK1 · 2021-08-31
> **Read**
>
> Thank you for your detailed answers, those clarify most of my doubts.
> I think there's merit in this work, the main limitation I see is the restriction of only dealing with two-player zero-sum games.

---

### Official Review · Reviewer_kzxK · 2021-07-10

**Rating:** 5
**Confidence:** 4

**Summary:**

The paper presents a new approach for learning to play against an opposing agent in a two-player zero-sum game that can transition among an arbitrary set of strategies. Experiments in two-player Kuhn poker demonstrate that the approach obtains high degrees of exploitation and low values of exploitability compared to several benchmark approaches.



**Limitations And Societal Impact:**

I don't see any potential negative societal impact of the work.

**Main Review:**

There appears to be a major problem with the paper. The main algorithm uses an RNR solver based on an LP formulation that is provided in the appendix, but this formulation is incorrect.

In particular, the first LP on page 579 on page 14 is incorrect, since the vector y_a is defined incorrectly on line 578. It should be y_a(j) = p pi_fix(a)  for j != a, not y_a(j) = 0 for j != a (the j=a case is correct).

There also should not be a need for the second LP, since player 1’s strategy should correspond to the dual variables from the first LP.

The second LP does appear correct, though the second set of constraints (pi^2(a) <= p pi_fix(a) + 1 – p) is redundant and implied by the final two constraints.

Since the algorithms are based on an incorrect formulation, I assume all the experiments are incorrect, so I can’t really assess them further.

I think that overall the problem is important and the approach is promising.

Some other minor comments:
37: dominated actions are not the only forms of irrationality in games. In fact, dominated actions are very rare in realistic poker games. See “Mistakes in Games” Ganzfried DAI 2019.

38: a NE -> an NE

naive -> naïve

90: Seems weird to cite a recent 2020 paper for the result that two-player zero-sum games can be solved in polynomial time by an LP.

109: States that the approaches can be extended to multiplayer games, but RNR is defined only for two-player zero-sum games, so it should be elaborated on how the approach is applicable to multiplayer games.

Section 3.1 seems to just present the standard definition of a normal-form game. I’m not sure why it is called a “meta normal-form game.” What is the “meta” part referring to that differentiates from a standard NFG?

Footnote i refers to stochastic games, while the model in the paper is just for normal-form games. Why is there a reference to stochastic games here?

The game model described is for normal-form games, but experiments are on Kuhn poker, which is an extensive-form imperfect-information game.

152: life-time -> lifetime

156: why use a maximum-a-posteriori approach instead of alternatives?

163: rolls-out -> rolls out

It would be helpful if some more information about the PPO algorithm is included, since it is the opposing agent as well as one of the benchmark algorithms.

I think it might be helpful to include a static Nash equilibrium strategy as an additional benchmark strategy (there are multiple equilibria in this game, so perhaps some of the extreme point NE can be used).

Does “stds” mean “standard deviations”?

In Table 1 the standard deviations seem very large compared to the payoff values, which indicate that additional experiments should be run.

Additional comments after the responses and discussion:

Based on the response it is likely that the authors probably used the correct LP formulation in the experiments and just wrote it wrong in the paper.

The authors claim that they have done additional experiments that achieve the same result. I'm not sure if we can just take their word. I think it is necessary to obtain statistical significance, and the current results do not come near this. In the response to 8Pgk they state that they also have comparisons against other benchmark agents SAM and MCCFR that show superiority of the new approach. But again I'm not sure if we can take into account results not in the paper. I would lean towards marginal acceptance if results showing statistical significance as well as comparisons against SAM and MCCFR were included.

I would still be interested in seeing a comparison against static NE strategies (there are multiple NE). I think this is a very important benchmark to include.

Author response:
"Mistakes in Games (Ganzfried 2019): Thank you for pointing out this interesting work. We will reword “dominated actions” to “dominated strategies” and discuss the difference between the two terms with reference to the work."

That paper actually shows that both dominated actions and strategies are very rare in realistic poker games, and proposes a new concept of a "mistake" that is more prevalent.





**Time Spent Reviewing:**

6

---

> ### Author Response · Authors · 2021-08-10
> **Comments for Reviewer 3**
>
> Thank you for taking time to review our work and providing detailed and insightful feedback.
>
> Addressing your questions:
> 1. LP formulation: We agree that y_a(j) = p x pi_fix(a) for j!= a rather than y_a(j) = 0 for  j!= a. However, this is a typo in writing our appendix rather than a mistake when we formulate and implement the RNR solver by LP. There are several ways to prove that it is indeed a typo: 1. Our experiment's results do show that our agent has the RNR solution’s properties where the agent can play an approximate nash strategy when it has low confidence about its opponent model’s prediction and plays an approximate best response to the prediction otherwise. 2.  We derive the y_a vector from the inequality below line 576 in the appendix. If the error comes from our misunderstanding of the LP formulation, this inequality relationship should also be incorrect. This inequality relationship being correct also proves the error is a typo. 3. After the whole review process is completed and the authors’ identities are revealed to the public, we can make our repository open to the public and leave its link in the updated version of our paper. Then you could check the git commit history and see we implemented the solver correctly before submitting the paper. We do apologise for making a typo in the formulation and hinder your understanding of our work. However, the idea of applying DP to model non-stationary opponents and combining RNR solution, PSRO style training with this opponent model is sound and novel. We hope you could take into account that the error is a typo and our idea is well-motivated and solid and reevaluate our paper.
>
> 2. Mistakes in Games (Ganzfried 2019): Thank you for pointing out this interesting work. We will reword “dominated actions” to “dominated strategies” and discuss the difference between the two terms with reference to the work.
>
> 3. Extension to multiple players: As we rely on DP-based opponent models to model non-stationary opponent’s policy, we can effectively regard multiple opponents as one opponent and use our opponent model to model the non-stationary joint opponents’ policies. RNR works on the policy space level. Therefore, on the policy space level, our problem is still a two-player zero-sum game from our agent’s perspective.
>
> 4. Meta normal-form game: The definition is effectively a normal form game. We name it meta normal-form game to emphasize this is defined over policy space where each pure strategy in the meta normal-form game is a (stochastic or deterministic) policy in the low-level game. The word “meta” referring to policy-space level is also used in [21]. We assume the actual game players play against each other can be formulated as a stochastic game, therefore we also include stochastic game’s definition in our appendix.
>
> 5. Maximum-a-posteriori approach: As explained in section 4 (line 200-204), we use a sequential maximum-a-posterior approach for the policy indexes $z_{1:m}$ due to the following reasons: (1) It prevents creating a new policy at each step when the latest trajectory is hard assigned to an existing policy; (2) It requires updating only one policy at each step. This approach significantly reduces memory usage and accelerates policy updating (M-step).
>
> 6. Including static NE as a benchmark: We will consider adding this as an extra benchmark in our next version. However, we have the concern it may not add extra information related to the focus of our work.
>
> 7. Details of PPO: We have included all critical hyper-parameters of our PPO implementation in Appendix Table 2 and the detailed algorithm could be found in [37].
>
> 8. Stds and more random seeds: “Stds” means “standard deviations”. We have rerun all experiments whose results are presented in Figure 2 and Table 1 with extra 20 random seeds. Our conclusions remain the same with these extra experiments. The values in our figures and tables will be updated in our next version accordingly.

---

> ### Author Response · Authors · 2021-08-27
> **New experiments results**
>
> Thank you very much for trusting us that the error is a typo rather than a formulation error.
>
> For your concern about the statistical significance of our methods compared to other baselines, please see our latest comment to all reviewers titled "Statistical Significant Performance".
>
> For static NE strategies baselines, we create one NE strategy as follow:
>
> Given a parameter \alpha in [0, 1/3], Player one chooses to bet with probability \alpha when he has a Jack and checks otherwise. Then, when the other player bets he will always fold. Player one chooses to bet with probability 3*\alpha when he has a King and checks otherwise. Then, when the other player bets he will always call. Player one will always check when he has a Queen. Then, when the other player bets after the check, he will call the probability (1/3+\alpha) and fold otherwise. The second player has a single equilibrium strategy: He will always bet or call when he has a King. He will check if possible when he has a Queen, otherwise, he will call with the probability of 1/3. When having a Jack, he never calls and bets with the probability of 1/3.
>
> Given the rule above, we create 4 sets of NE strategies by setting \alpha equal to 0, 1/5, 1/4, 1/3 respectively. We present our results below where values in parentheses are stds taken over 20 random seeds.
>
> For \alpha = 0, the NE strategy can obtain exploitation to PPO, TRPO, A2C opponents at 0.038 (0.006), 0.028 (0.009), 0.056 (0.014) respectively.
>
> For \alpha = 0.2, the NE strategy can obtain exploitation to PPO, TRPO, A2C opponents at 0.048 (0.011), 0.030 (0.008), 0.090 (0.057) respectively.
>
> For \alpha = 0.25, the NE strategy can obtain exploitation to PPO, TRPO, A2C opponents at 0.045 (0.009), 0.031 (0.008), 0.084 (0.037) respectively.
>
> For \alpha = 1/3, the NE strategy can obtain exploitation to PPO, TRPO, A2C opponents at 0.038 (0.006), 0.022 (0.005), 0.068 (0.040) respectively.

---

### Official Review · Reviewer_8Pgk · 2021-07-14

**Rating:** 3
**Confidence:** 4

**Summary:**

The paper proposes to use the Dirichlet process for opponent modelling in games. It argues that the approach is particularly suitable for non-stationary opponents. It then proposes an algorithm to compute a counter-strategy to the model based on the restricted Nash response. Empirical evaluation on Kuhn poker shows that this approach can exploit a non-stationary learning opponent similarly to one of pre-existing methods, while maintaining relatively slow exploitability.

**Limitations And Societal Impact:**

There is not much discussion on limitations or societal impact. Thorough discussion of computational costs would certainly be an improvement. I do not believe the paper has substantial societal impact.

**Main Review:**

Originality: The problem of opponent modeling in games has been extensively studied before. The idea to use the Dirichlet process (DP) to solve it may be new and is well motivated in the paper.

Clarity: The paper is reasonably clear and understandable. However, it would be further improved by use of examples to demonstrate the notions and algorithms on a specific tiny game.

Quality: The notation and basic claims in the paper seem sound. My main concern is with the positioning with respect to the related work and the quality of the empirical evaluation.

(1) Related work. Opponent modelling in games is a classical problem with many solutions. This paper reviews a part of this literature, but it does not provide theoretical or empirical arguments on superiority of the proposed approach to the existing ones. A natural candidate for the context of the paper would be, for example, the implicit agent modelling framework [A], but at least a compelling argument of superiority to [10,15,49] either theoretically or empirically should be provided.

(2) Computational complexity. There is no discussion of the computational complexity of the proposed method. It looks much more expensive than some of the alternatives mentioned above and the evaluation is only on a tiny game. How well scalable would the approach be to larger games?

(3) Experimental evaluation. The experiments are performed on one small game against one opponent on 5 runs of the system, without any statistical analysis of the confidence intervals. It is not sufficient to make any conclusions.

Significance: The proposed approach might be a good idea. However, without a clear comparison to competing methods, I do not think someone would choose to implement and use it on some practical problem. From the theoretical perspective, I understand the paper as an application of an existing method of DP to the problem of opponent modelling. Hence, there is no theoretical advancement. If it is not so, I assume the authors will argue for the theoretical novelty in the rebuttal.



References:

[A] Bard, N., Johanson, M., Burch, N., & Bowling, M. (2013, May). Online implicit agent modelling. In Proceedings of the 2013 international conference on Autonomous agents and multi-agent systems (pp. 255-262).


**Time Spent Reviewing:**

3

---

> ### Author Response · Authors · 2021-08-10
> **Comments for Reviewer 2**
>
> Thank you for taking time to review our work and providing detailed and insightful feedback.
>
> Addressing your questions:
>
> 1. Relation to Bard et al. (2013): In Bard’s work, they observed that explicit opponent modelling requires a large amount of computation and training data, which makes the online opponent modelling challenging. Therefore, they proposed to move the heavy computation offline and only perform light online inference by implicit opponent modelling. Specifically, they built a population of robust response policies offline by RNR and DNR methods and only infer the expected utilities given its opponent’s behaviour and response policies online. Then they can do online adaptation via multi-armed bandit algorithms with online inferred expected utilities and a group of experts trained offline. However, as with previous works related to RNR, Bard did not consider the non-stationarity induced by adaptive opponents. For example, the opponent the agent encounters during the online test stage is still a trained opponent with a stationary policy. We know that value-based RL methods have no theoretical guarantee of convergence in non-stationary environments and often perform badly in non-stationary experiments too. Therefore, we think Bard’s work may also suffer the same problem as they also rely on opponents being stationary so that they can have a sufficiently accurate estimation of the expected utilities of a portfolio of counter strategies. We appreciate you pointing out the relation of Bard’s work to CEPSOM and will include the above discussion in our updated version.
>
> 2. New baselines: We implemented SAM [10] and MCCFR[B] as other two baselines. Similar to BC, SAM has high exploitability when it is trained against a PPO opponent. Its exploitation to the PPO agent also varies greatly. Once a SAM agent is trained, we test it against new adaptive opponents. When facing PPO, TRPO and A2C agents, it can only achieve -0.271 (0.167), -0.298 (0.105), -0.259 (0.171) exploitation respectively. This is slightly better than normal BC as SAM can effectively switch between two opponent models but is still very limited. Because all other baselines have no direct access to their opponent’s policy, to have a relatively fair comparison, we adopt the Monte Carlo sampling version of CFR. However, even in this setting, MCCFR still has advantages as it can query its opponent to sample actions for its update which is not allowed in our other experiments. We did not consider a self-play version of CFR as we know it has similar performance to PSRO, i.e. converge to Nash equilibrium strategy, so it will not add extra information. MCCFR trained against adaptive PPO can reduce its exploitability to a relatively low level (0.28) but is limited to exploit the PPO opponent, especially considering that we only evaluate how much a baseline can exploit the PPO opponent after every time the PPO is updated. We also test trained MCCFR against new adaptive opponents including new PPO, TRPO, and A2C and it can only achieve exploitation of  -0.171 (0.162), -0.106 (0.108), 0.08 (0.141) respectively. We will update Figure 2 including the learning dynamics of these two new baselines against PPO opponents in our paper.
>
> 3. Computational complexity: The computational complexity of our algorithm can be considered in three parts: 1. learning of an opponent model 2. learning of a best response policy to a prediction by the opponent model 3. solving the meta-game for RNR solutions by linear programming. In complex environments, to have relatively accurate models, the computational complexity of the first two parts is mainly affected by the number of data points we used during the training and is polynomial of the number of data points. Solving RNR solutions by a linear program (LP) is in polynomial time of the meta-game size. As done in [A], we could move most computationally heavy parts (1 and 2) offline. When we need to play against non-stationary opponents at test time, we would then only need to do light opponent inference and solve RNR solutions for the meta-level game online.
>
> 4. Experimental evaluation: We apologise that we do not fully understand your question. You mentioned that “...without any statistical analysis of the confidence intervals”. We believe presenting empirical results with standard deviations of sample data obtained from different random seeds is a common approach used in the current machine learning community. Therefore, it may not be fair to say we do not have any statistical analysis because we do not present confidence intervals. If you were to mean that 5 random seeds are not enough, we have rerun all experiments whose results presented in Figure 2 and Table 1 with extra 20 random seeds and our conclusions drawn from these results remain the same as before. The values in our figures and tables will be updated accordingly in the next version of our paper with these 20 random seeds. We admit our test environment is very simple, however, we think having an algorithm to work well on the problem we focused on is not easy. Our problem is challenging because 1. We can only observe opponent history trajectories and knowledge of the opponent's learning characteristics is very limited. 2. We not only require our algorithm to learn a good approximate nash strategy but also be able to exploit a set of sub-optimal opponents much better than the Nash strategy. 3.Our agent does not need extra training at test time. However, opponents during training and test time are both allowed to learn and adapt and opponents encountered in test time can use different learning algorithms from opponents’ in training time 4. We only evaluate our agent exploitation to an opponent every time after the opponent just updates its policy where we often observe that the exploitation is lowest in the period between which our agent just updates and the opponent just updates. To the best of our knowledge, we do not see many works satisfying all the above requirements.
>
> [B] Marc Lanctot, Kevin Waugh, Martin Zinkevich, and Michael Bowling. 2009. Monte Carlo sampling for regret minimization in extensive games. In Proceedings of the 22nd International Conference on Neural Information Processing Systems (NIPS'09). Curran Associates Inc., Red Hook, NY, USA, 1078–1086.

---

> > ### Comment · Reviewer_8Pgk · 2021-08-24
> > **The response did not change my opinion**
> >
> > Thank you for your rebuttal, but it unfortunately did not change my mind.
> >
> > (1) The previous work is based on building a portfolio of strategies and using an *adversarial bandit* algorithm on top of it in online play. I think it makes it very suitable for non-stationary environments.
> > (2) I believe that Bart et al. is really a necessary baseline to thoroughly compare to and other baselines cannot substitute it.
> > (3) I do not fully understand how you mean the offline pre-computation and it sounds quite different from what is currently written in the paper. I still think the presented approach is likely to be prohibitively expensive for most practical situations.
> > (4) 20 runs may help to make the results significant, however, if we look at the current Figure 2, there are likely differences in exploitability of the approaches, but the exploitation is one big mess with the shading with undefined meaning completely overlapping and I do not see how we can make any conclusions from those results. Similarly in Table 1, the reported STDs are rather large and you just compare the mean without any statistical test about the significance of the results. With 5 samples, the STD roughly corresponds to the 95% confidence intervals and in Table 1, it means that you most likely cannot claim that your methods are significantly better than PSRO.

---

> > > ### Author Response · Authors · 2021-08-25
> > > **Looking forward to hearing from you**
> > >
> > > Thank you for your response. We will respond to your further comments below. In the meantime, we also want to have some clarifications from you.
> > >
> > > (1) First, we want to clarify again what we mean by a non-stationary environment. In our setting, this mainly refers that you will encounter opponents who can learn and adapt to your behaviour while you playing against it. For example, the opponent learns by a PPO algorithm while playing against you in our paper. Unfortunately, we did not see any theoretical discussion about the non-stationary environment in Bard’s work (Bard et al. (2013)): 1. We re-read the paper [C] which proposed the method used for utility estimation in Bard et al. (2013). However, they assume a stationary environment. 2. In Bard et al. (2013), they denote “the expected utilities of a portfolio of counter strategies” as u_{i}(\sigma_{-i}, \sigma^{A}_{i}). No super/subscripts are given to the opponent’s strategy \sigma_{-i} indicating time step, which also shows that Bard et al. (2013) assumes stationary opponent. 3. They used EXP4 as their multi-arm bandit algorithm. However, “Existing MAB ensembles are not robust to non-stationarity” (from [D]). “Given such an oracle, however, it is known that no efficient low regret algorithms exist in the fully adversarial setting (Hazan and Koren, 2016, Theorem 25), even without any challenges of non-stationarity. Consequently all previous works explicitly rely on assumptions such as i.i.d. contexts, or even i.i.d. context-reward pairs.” (from [E]). In addition, they did not use any of their empirical results to claim “ it makes it very suitable for non-stationary environments”.
> > >
> > > We explained why we did not consider Bard et al. (2013) as a baseline in our rebuttal because of its limitation. However, you ignored all our extra results requested by you and strongly suggested we include Bard et al. (2013). Therefore, we would like to know:  1.how do you make this claim “I think it makes it very suitable for non-stationary environments.” from Bard et al. (2013)? 2. What evidence from the paper you found supporting the claim? 3. How strong the evidence is?
> > >
> > > (2) Based on your previous comment “but at least a compelling argument of superiority to [10,15,49] either theoretically or empirically should be provided”, we implemented another two baselines SAM and MCCFR from the list you provide. SAM specifically claimed in their paper they focus on the non-stationary setting as we do. We show our algorithm can outperform both baselines. However, you ignore all of our extra efforts to respond to your request and explain it as “I believe that Bart et al. is really a necessary baseline to thoroughly compare to and other baselines cannot substitute it.”
> > > Therefore, we would like to know why did you propose “at least …. [10,15,49].”? It seems these two comments from you contradict each other.
> > >
> > > (3) In Table 1 we show once trained, our algorithms (EPSOM/CEPSOM) can still play well against new adaptive opponents compared to other baselines. These results verified that we can train our EPSOM/CEPSOM offline to build a population of policies and modelled opponent policies. Then use the trained agent to play against new opponents online.
> > >
> > > (4) It is natural and expected to see large fluctuations of performance in our case. This is because: 1. all adaptive opponents in our work are SARL algorithms, the initializations of their parameters and random seeds greatly affect their learning process. Therefore, the randomness is also reflected in the performance. 2. Because opponents are adaptive and learning during the tournaments, there would be times when our agent is confident to exploit them and times when our agent is unconfident and play a safe strategy to avoid potential exploitation by the opponents. Therefore, the variance/stds of the exploitation are quite high. Let’s consider a simple and illustrative example: let us assume there are two players playing shooting games. Player A’s scores are [1, 3, 5, 7, 9] over five shootings and Player B’s scores are [3, 3, 3, 3, 3]. The average and std of A’s scores are 5.0 and 2.83 respectively. The average and std of B’s scores are 3.0 and 0 respectively. 5-2.83=2.17 which is smaller than B’s average score. Even though A outperforms B by a very large margin 3 times out of 5 shootings and ties with B 1 time out of 5 shootings, we cannot say anything about the relative strength of A compared to B by your reasoning. This example is close to our case because when the opponents are exploitable and EPSOM/CEPSOM are confident about it, EPSOM/CEPSOM can exploit the opponents to a very large extent, leading to high exploitation. When opponents are NOT exploitable or EPSOM/CEPSOM are NOT confident about it, EPSOM/CEPSOM play a safe strategy that has low exploitation. Most of the time our agent obtain positive exploitation and the large fluctuation comes from large and positive extreme values.
> > >
> > > We spent lots of time on the paper and rebuttal based on your comments, so we expect fair review. We hope our new comments will clarify some of your questions and look forward to hearing from you.
> > >
> > > [C] M. Bowling, M. Johanson, N. Burch, and D. Szafron. Strategy evaluation in extensive games with importance sampling. In Proc. of the 25th Annual Int. Conf. on Machine Learning (ICML), 2008.
> > >
> > > [D] Pang, K., Dong, M., Wu, Y. and Hospedales, T.M., 2018, August. Dynamic ensemble active learning: A non-stationary bandit with expert advice. In 2018 24th International Conference on Pattern Recognition (ICPR) (pp. 2269-2276). IEEE.
> > >
> > > [E] Luo, H., Wei, C.Y., Agarwal, A. and Langford, J., 2018, July. Efficient contextual bandits in non-stationary worlds. In Conference On Learning Theory (pp. 1739-1776). PMLR.

---

> > > > ### Comment · Reviewer_8Pgk · 2021-08-31
> > > > **Quick clarification**
> > > >
> > > > Hello
> > > >
> > > > (1) The method is based on adversarial bandit algorithm, which is guaranteed to achieve vanishing regret against any sequence of opponent strategies. Therefore, I think it is particularly suitable for non-stationary opponents. If you want even nicer guarantees, you can use a bandit with a tracking regret guarantee, which is a trivial modification of the existing method.
> > > >
> > > > (4) I understand the sources of variance very well, but absolutely disagree that if we have high variance, we can just ignore statistical significance and present results that can very well be random noise. In your example, you absolutely cannot say anything about whether A or B is better without additional assumptions.

---

> > > > > ### Author Response · Authors · 2021-09-02
> > > > > **Thank you for your reviewing**
> > > > >
> > > > > Dear Reviewer
> > > > >
> > > > > Thank you for your response. However, we still do not see the strong necessity that Bard et al. (2013) is “ really a necessary baseline to thoroughly compare to and other baselines cannot substitute it.” from your latest comment. Reasons are listed below:
> > > > >
> > > > > 1. You did not clarify how to estimate the utilities of the portfolio of the response strategies given an adaptive opponent. The paper [C] which proposed the method used for utility estimation in Bard et al. (2013) assumes a stationary environment.
> > > > >
> > > > > 2. We understand EXP4 can achieve sublinear regret in an oblivious adversary (adversary chooses the rewards/cost at the start of the game [F]) case. However, in our setting, opponents adapt to the training agent’s behaviour during the interactions. Therefore, they are non-oblivious adversaries. “The standard notion of regret losses much of its meaning”[G] in this case.
> > > > >
> > > > > 3. You did not provide any direct theoretical or empirical proof or quote from Bard et al. (2013) supporting your claim that “ it makes it very suitable for non-stationary environments.”
> > > > >
> > > > > For the statistical significance issue, please see the new comment with the title “Statistical Significant Performance” to all reviewers. We recalculated the stds of our algorithms and baselines so that the overall performances of these methods can be compared in the standard way.
> > > > >
> > > > > [F] Lattimore, T., & Szepesvári, C. (2020). Bandit Algorithms. Cambridge: Cambridge University Press. doi:10.1017/9781108571401
> > > > > [G] Arora, R., Dekel, O., & Tewari, A. (2012). Online bandit learning against an adaptive adversary: from regret to policy regret. arXiv preprint arXiv:1206.6400.

---

> > > > > > ### Comment · Reviewer_8Pgk · 2021-09-06
> > > > > > **More clarification would be IMO needed**
> > > > > >
> > > > > > Dear authors,
> > > > > >
> > > > > >   I am sorry, but I am still not convinced by your arguments. I do not see any assumption of stationarity in the value estimation, since it is estimating the expected utility of playing a fixed strategy in a single play-out of the game. Also, I do not see why the opponent would not be oblivious to the randomisation in the bandit algorithm and even with such a super-strong adversary, I do not see how and why it would be handled better by the proposed approach. I am not saying it is not true, but I would certainly prefer to see and review the updated paper that properly explains it before accepting it.

---

> > > > > > > ### Author Response · Authors · 2021-09-07
> > > > > > > **Thank you for your reviewing**
> > > > > > >
> > > > > > > Dear Reviewer,
> > > > > > >
> > > > > > > Thank you for your time.
> > > > > > >
> > > > > > > 1. As you said in the last comment, the method in [C] is " is estimating the expected utility of playing a fixed strategy". The transition probability $f_{c}$ associated with the chance node is also fixed (but unknown). Therefore, the environment in [C] is stationary, i.e. the dynamics of the environment and the interactions of agents are stochastic but consistent over time. However, in our setting, the opponent is adaptive. Its policy will change over time, i.e. neither fixed nor known. Therefore, our problem is non-stationary.
> > > > > > >
> > > > > > > 2. As our opponent can adapt to the training agent's behaviour, its policy $\pi^{-i}_{t}$ is effectively a function of the history of the training agent's policies. Therefore, the adaptive opponent is non-oblivious.
> > > > > > >
> > > > > > > Based on all our comments, we believe you are giving us very unfair reviews:
> > > > > > > 1. You ignored all the merits in our paper or the new results we provided in the rebuttal.
> > > > > > >
> > > > > > >
> > > > > > > 2. You made a very subjective claim that [A] "is very suitable for non-stationary environments". We requested you to provide direct theoretical or empirical proof or quotes from [A] but you never respond to it.
> > > > > > >
> > > > > > > 3. Your comment "I do not see any assumption of stationarity in the value estimation, since it is estimating the expected utility of playing a fixed strategy in a single play-out of the game." is also wrong.
> > > > > > >
> > > > > > > 4. You cannot distinguish the difference between oblivious and non-oblivious adversaries.
> > > > > > >
> > > > > > > 5. Whenever we tried our best to clarify some points you have questions or concerns by patient discussion or new empirical results, you did not respond on those points anymore but only throwing out new questions or excuses for rejecting our paper.
> > > > > > >
> > > > > > > 6. For example, we ran new baselines requested by you but you completely ignored them only because we did not provide another baseline result. However, you failed to provide any strong or direct evidence from the baseline's original paper to prove that the baseline "is very suitable for non-stationary environments".
> > > > > > >
> > > > > > > 7. Another example: We tried to explain to your our adaptive opponent is non-oblivious, which is different from the common oblivious adversary setting. Then you throwing out the new excuse that " I do not see how and why it would be handled better by the proposed approach" to reject our paper. We do not see any other reviewers having the question about how our method handles the adaptive opponent. More importantly, if you have this general question about our paper, why did you not ask it in the first comment but only waited till now?
> > > > > > >
> > > > > > > We are always happy to communicate with reviewers who are fair and qualified. But we can never change a reviewer's mind when he/she has decided to reject us at the beginning of the rebuttal regardless of what we do during the rebuttal.  Therefore, we decided not to respond to your comments any more.

---

> > > > > > > > ### Comment · Reviewer_8Pgk · 2021-09-07
> > > > > > > > **...**
> > > > > > > >
> > > > > > > > Hi again,
> > > > > > > >
> > > > > > > >   1) You are confusing the players. The exploitative strategies in the portfolio are static, therefore, you can estimate their utility in one play-out regardless of what the opponent does. There is no assumption about the opponent at all. The opponent can play a different strategy on each round and it has no impact on the correctness of the utility estimates.
> > > > > > > >   2)  Unless you assume that the opponent has access to the random numbers generation, I am convinced that Exp4 will achieve no regret in practice, as indicated by many experiments with co-evolution of regret minimisers in games, which are apparently both adaptive. If you would want to have theoretical guarantees for arbitrary non-oblivious opponents, you would have to be a little more careful and use something like Exp3.P instead of plain Exp3 [W]. However, I consider this discussion largely irrelevant to whether to accept your paper in its current form, since your paper argues mainly by practical performance, which should be IMO statistically correctly compared to the state of the art method for the problem you tackle.
> > > > > > > >
> > > > > > > > [W] Bubeck, S., & Cesa-Bianchi, N. (2012). Regret analysis of stochastic and nonstochastic multi-armed bandit problems. arXiv preprint arXiv:1204.5721.

---

> > > > > > > > > ### Author Response · Authors · 2021-09-08
> > > > > > > > > **New comments**
> > > > > > > > >
> > > > > > > > > Dear Reviewer,
> > > > > > > > >
> > > > > > > > > Thank you for your time and patience in writing us new comments.
> > > > > > > > >
> > > > > > > > > 1. [A] is estimating the utility of a portfolio of strategies for the training agent. But one player's utility also depends on the opponent's strategy. The method in [C] focuses on estimating the expectation of one player's utility given the joint strategy. If the distribution is not stationary, the expectation is not a converging value at all. We have not been confused about the players or "confusing the players".
> > > > > > > > >
> > > > > > > > > 2. "you can estimate their utility in one play-out regardless of what the opponent does." If this was true, there would be no need for online utility estimation. The estimation of the portfolio should be done offline by your argument as you do not care about what the opponent does.
> > > > > > > > >
> > > > > > > > > 3. Furthermore, if the above argument was true, we believe the author would have an explicit discussion about this significant advantage in their original paper. Could you find any quotes from the original paper as we could not? We are afraid you might ignore this request as well because you never provided direct theoretical or empirical proof or quotes from [A] to support your claim that [A] "is very suitable for non-stationary environments".
> > > > > > > > >
> > > > > > > > > 4. "However, I consider this discussion largely irrelevant to whether to accept your paper in its current form". We never intended to have this long discussion on the point. It is only because you said, "I believe that Bart et al. is really a necessary baseline to thoroughly compare to and other baselines cannot substitute it. " and you ignored all our new results. Then we tried to explain to you why [A] is not that necessary.
> > > > > > > > >
> > > > > > > > > 5. For the statistical significance issue, please see our comment with the title "Statistical Significant Performance".
> > > > > > > > >
> > > > > > > > > 6. For the SOTA baseline, please note that
> > > > > > > > > (1). We can only observe opponent history trajectories and knowledge of the opponent's learning characteristics is very limited. (2). We not only require our algorithm to learn a good approximate nash strategy but also be able to exploit a set of sub-optimal opponents much better than the Nash strategy. (3). Our agent does not need extra training at test time. However, opponents during training and test time are both allowed to learn and adapt and opponents encountered in test time can use different learning algorithms from opponents’ in training time (4). We only evaluate our agent exploitation to an opponent every time after the opponent just updates its policy where we often observe that the exploitation is lowest in the period between which our agent just updates and the opponent just updates.  If any previous published SOTA also satisfying the above conditions, we would be happy to compare it with.

---

### Official Review · Reviewer_oaZz · 2021-07-16

**Rating:** 6
**Confidence:** 4

**Summary:**

This paper presents an algorithm for playing 2-player zero-sum repeated games that attempts to effectively balance the ability to model and exploit other agents while not opening one’s self to exploitation.  Limited results in Khun Poker when paired with three different algorithms are presented, with comparison to a handful of other methods.


**Limitations And Societal Impact:**

The paper doesn't explicitly address these aspects.  The limitations of the evaluation need to be further stated.  I don't think there is anything really to say about societal impact.

**Main Review:**

Initial reflection for context: The paper addresses an old problem that has, from my perspective, been re-hashed repeatedly over the last several decades.  Despite being foundational and somewhat simple, the answers remain extremely difficult and hard to come by (I argue that the problem itself is ill-posed).  It is within this view that I base my review.

Strengths:

+ The paper is, for the most part, well-written and polished.  The technical description of the algorithm is detailed and seems solid (albeit I found the algorithm description difficult to follow).

+ I really like the two measures the paper analyzes: ability to exploit and exposure to exploitation.  This trade-off is central to the problem at hand.  (I expound on this a bit in the weaknesses portion of the paper).

+ For the empirical results that are presented, the algorithm appears to perform well.  The results are limited, and in some sense can be viewed as simply anecdotal.  However, I also do not want to diminish the strength of this contribution, as getting an algorithm to do this even in this limited context is no small accomplishment.

Weaknesses:

- I read the results differently than how they are discussed in the text of the paper (which seems to be bent on advocating, perhaps understandably so, for the algorithm presented in the paper).  To me, the key insight of the paper is the trade-off between exploitability and exploitation.  I’m not sure what the ideal trade-off between these measures is, but we could (for example) weight them equally.   If we do that, Figure 2 clearly shows that exploitability has higher magnitude across the algorithms than does exploitation.  If we simply subtract exploitability from exploitation, it appears to me that only PSRO has a positive value, suggesting it may be the best algorithm in this case among the 5 presented (though, again, the trade-off assessment of equal value between the metrics is subjective).  In the second set of results shown in Table 1, a similar trend continues, except that CEPSOM also has positive weight.  But the results don’t seem to be telling me that CEPSOM necessarily obtains a better trade-off than PSRO (which is essentially minimax in this game, as I understand it).

- In all honesty, it is hard to make any general assessment about EPSOM and CEPSOM from these limited results.  Is there some opponent that could really exploit CEPSOM.  If so, when and how?  If not, when and how?  The authors state at the outset of the paper that they are not interested in self play.  But self play is a foundational comparison in this domain, as any good algorithm would be adopted by other agents over time necessitating a successful algorithm to be good (in some sense) in self play.  But modeling copies of one’s self is often problematic.

- I do not understand Figure 3.

Again, this topic has been addressed for decades now, and I don’t know that we are any closer now to solving the problem than we were decades ago.  Perhaps that is because the problem itself is, in a loose sense, ill-posed — claiming algorithms are better than each other in this domain with trade-offs are largely subjective.  That is why I like the set of metrics (exploitation and exploitability) present in this paper — it looks at the (messy) trade-off of the problem.  I wish the authors had taken that trade-off further in their analysis.  If they are will to do that, I can envision the paper as a worthy contribution.


**Time Spent Reviewing:**

?

---

> ### Author Response · Authors · 2021-08-10
> **Comments for Reviewer 1**
>
> Thank you for taking time to review our work and providing detailed and insightful feedback. We are glad that you found the underlying idea behind EPSOM interesting and understand how challenging the problem is. We believe the trade-off between exploitation and exploitability is an important topic in competitive multi-agent problems. Many prior works [3,9,18,19,21] have also studied this topic so we cannot take all the credit for providing the trade-off analysis. Our contribution to this topic is that we take into account the non-stationarity induced by opponents when we try to optimise for this trade-off. Most prior works only consider how to exploit one stationary opponent while keeping the agent itself with low exploitability. However, in realistic settings, we need to consider how to exploit opponents which change over time while maintaining low exploitability. Changes can come from opponents also having the ability to learn and adapt or opponents that are different in different tournaments or both. Therefore, our work extends previous works to the non-stationarity setting which is more realistic and therefore more significant to the progress of research in this area.
>
> Addressing your questions:
>
> 1. Interpretation of our results: In our experiments, we mainly want to show that (C)EPSOM can exploit non-stationary opponents while maintaining a low exploitability policy. In Figure 2 we can see that (C)EPSOM achieves similar exploitation when facing an adaptive PPO opponent as BC does but without sacrificing exploitability. In Table 1 we show that, once trained, (C)EPSOM can still exploit other adaptive opponents different from the ones (C)EPSOM encountered during the training without any further training.
> 2. Difference between exploitation and exploitability: The exploitation is specific to a given opponent and measures how much a policy can exploit the given opponent. The exploitability only considers the worst case and measures how exploitable a policy is. Firstly, given an opponent, the difference between the two terms lacks a straightforward interpretable meaning. Secondly, if we weigh the two terms equally, the difference cannot tell us generally how good an algorithm is. When facing an opponent with a randomly initialised policy, EPSOM, CEPSOM and PSRO can achieve exploitation of 0.668, 0.739 and 0.276 respectively. When we subtract their exploitability reported in Table 1 from the above exploitation, we can see (C)EPSOM has significant advantages over PSRO. However, a randomly initialised opponent is rare in real tournaments. Therefore, if we try to calculate the difference between them, we should take the expectation of these terms with respect to the probability that we will encounter the given opponent and the worst-case opponent respectively. Then this requires us first to define the sample space. Focusing on a specific tournament with a limited number of opponents or the whole opponent policy space can greatly affect the result. Therefore, how to correctly calculate the difference between exploitation and exploitability is not an easy problem and beyond the focus of our work.
>
> 3. Is CEPSOM exploitable by other opponents?: As we can see from Table 1, the lowest exploitability CEPSOM can obtain is 0.08. This value is positive which means CEPSOM is exploitable. However, to truly exploit CEPSOM, one must disable CEPSOM’s ability to adjust its meta-strategy according to its opponent model’s prediction. In this case, CEPSOM must then play the safest strategy (one with the lowest exploitability). Otherwise, when an opponent tries to exploit CEPSOM it will render itself exploitable too. CEPSOM may then spot its weakness and in turn exploit the opponent. Due to CEPSOM’s flexibility to adjust its meta-strategy, it is not easy for an opponent to exploit it for a long time.
>
> 4. Better trade-off than PSRO: Trained (C)EPSOM maintains a population of policies and an opponent model which can predict the encountered opponent and adjust its response strategy based on the prediction and the prediction uncertainty. To empirically assess (C)EPSOM's performance, we select a set of typical strong opponents such as PPO, TRPO and A2C (added in the updated paper following review) and weak opponents such as a randomly initialised opponent. (C)EPSOM can exploit all of them to a greater extent than PSRO while only having slightly higher exploitability. The advantage of (C)EPSOM against one type of opponent over PSRO may not be significant but the total benefit will accumulate when an agent faces an increasing diversity of opponents while our (C)EPSOM’s exploitability remains low and constant. Therefore, we believe (C)EPSOM has a better trade-off than PSRO.

---

### Author Response · Authors · 2021-08-10
**General Comments**

Thank you all for your detailed and insightful comments. We wish to resolve as many of your questions as we can in this rebuttal and improve our paper based on these valuable points. Minor issues such as notation, typos and rewording of some unclear statements will be corrected in our updated version and not discussed here.

Addressing some shared questions by multiple reviewers:
1. Self-play problem: the learning objective of many self-play algorithms in multi-agent problems is the convergence to Nash equilibrium. This essentially forgoes the exploitation of opponents and purely focuses on low exploitability. Contrastingly, our work focuses on training an agent with relatively low exploitability and a strong ability to exploit non-stationary opponents. Therefore, we state that “our work is not limited to self-play”. Currently, we focus on finding a solution with a good balance between exploitation and exploitability in a non-stationary setting. Therefore, opponents learning by SARL algorithm will suffice the requirement. If we consider any opponents which can also do opponent modelling (including (C)EPSOM), this will bring extra complexity such as recursive reasoning which is beyond our focus. Hence we do not intend to train EPSOM or CEPSOM by self-play.  We realise that our original statement may cause confusion. Therefore, we will simply state that our work is not a self-play algorithm in our updated version.

2. Figure 3: In this figure, we want to visualise and compare an RL agent's learning process in stationary and non-stationary environments respectively. To this end, we first create a stationary environment by fixing our trained agent CEPSOM to always play a Nash equilibrium strategy. Then from the perspective of the PPO agent, the environment becomes stationary. To create a non-stationary environment, we allow the trained CEPSOM to adjust its meta-strategy online based on the prediction of its opponent model. Creating a non-stationary opponent this way also allows us to understand how CEPSOM’s online adaptation affects the opponent’s learning process. To visualise the learning process, we rely on our opponent model. During the PPO agent’s training, our opponent model will predict its type based on recent trajectories produced by PPO. Then we use the t-SNE method to present these predictions in a 2-D diagram. As we can see from Figure 3, when the environment is stationary, PPO can learn and gradually converge towards the top side of the plot (right in Figure 3) However, when the environment is non-stationary, because CEPSOM adapts online, the PPO agent struggles to converge and bounce between the top and middle part of the plot (left in Figure 3).

---

### Author Response · Authors · 2021-08-27
**Statistical Significant Performance**

We have received comments concerning the statistical significance of our algorithms’ performance from multiple reviewers. We wish to use this comment to explain how we calculate the standard deviations (stds) in our paper, why their values are very large and what should be changed if we only want to compare our algorithm’s overall performance to other baselines:

1. In Table 1 from the original paper: For each trained agent (EPSOM, CEPSOM and other baselines), we host a tournament where the agent plays against an adaptive opponent (e.g. PPO) who can learn and update its policy during the whole tournament while the agent is not trained anymore in the tournament. Every time when the opponent updates its policy, we calculate the exploitation of the trained agent’s policy to the opponent’s updated policy. We stop the tournament when the opponent has updated its policy 100 times. We repeat the process 5 times over different random seeds. Then, we have 500 exploitation samples of one algorithm to an adaptive SARL algorithm. The mean and stds of one algorithm against a SARL algorithm in Table 1 are calculated over these 500 samples directly.

2. Why large stds: The main reason that the stds calculated this way are large is that the randomness of the opponent implemented by a SARL algorithm is also added into the stds. For example, the learning process of a PPO opponent with a policy parameterised by a neural network can be very different over different random seeds. In addition, given one random seed, a PPO’s policy can also be very different at its different learning steps (100 learning steps in total for one random seed in the paper). Therefore, if we calculate the stds over the 500 samples directly, we will have very large stds due to the randomness of the opponent’s learning process.

3. New stds are calculated: Therefore, if we want to compare the overall performance of C/EPSOM to other baselines, we should not consider the randomness induced by the opponent’s learning process and focus on the average exploitation to the opponent obtained by the algorithm over the opponent’s learning process. To achieve that, we rerun our experiments as follows: For one algorithm and its adaptive opponent, we calculate the average exploitation of the algorithm to its opponent over the opponent’s whole learning process (100 learning steps). We repeat the above process 20 times over different random seeds. Then we have 20 samples of the average exploitation of the algorithm to its opponent over the opponent’s learning process. Then we calculate the mean and stds over these 20 samples.

We will list our new results below, values in parentheses are stds taken over 20 random seeds:

For PPO opponents: EPSOM, CEPSOM, BC, PSRO, PPO, SAM, MCCFR can achieve average exploitation to the opponent over the opponent’s learning process at 0.037 (0.042), 0.114 (0.022), -0.562(0.011), 0.050 (0.014), -0.405 (0.014), -0.270 (0.003), -0.154 (0.023).

For TRPO opponents: EPSOM, CEPSOM, BC, PSRO, PPO, SAM, MCCFR can achieve average exploitation to the opponent over the opponent’s learning process at 0.097 (0.061), 0.115 (0.037), -0.276(0.126), 0.030 (0.006), -0.358 (0.031), -0.135 (0.010), -0.138 (0.027).

For A2C opponents: EPSOM, CEPSOM, BC, PSRO, PPO, SAM, MCCFR can achieve average exploitation to the opponent over the opponent’s learning process at 0.187 (0.075), 0.187 (0.029), -0.148(0.171), 0.086 (0.042), -0.347 (0.036), -0.107 (0.009), 0.115 (0.068).

Note that we did not change anything about the C/EPSOM algorithms or other baselines, but only how we calculate mean and stds in these new experiments in order to remove the randomness caused by the opponent's learning process.

---

### Decision · Program_Chairs · 2021-09-27

**Decision:**

Reject

**Comment:**

There was a detailed discussion about this paper, and the relationship to Bard et al. was controversial, with some reviewers considering this a necessary benchmark and others agreeing with the authors that it isn't (necessarily).  Since it's controversial, I'm inclined to not make the decision based on the Bard et al. concern (though I encourage the authors to make sure to have really thought this through before a next submission).  Even so, the overall level of excitement among the reviewers is a bit low for acceptance, lacking any real advocate for the paper.  The additional experiments provided by the authors are appreciated and will be good to include in detail in a future version.

I'll mark the paper as "reject but could be bumped up to accept" though I understand this means it's very likely to be rejected.